# Retinal Genomic Fabric Remodeling after Optic Nerve Injury

**DOI:** 10.3390/genes12030403

**Published:** 2021-03-11

**Authors:** Pedro Henrique Victorino, Camila Marra, Dumitru Andrei Iacobas, Sanda Iacobas, David C. Spray, Rafael Linden, Daniel Adesse, Hilda Petrs-Silva

**Affiliations:** 1Laboratório de Neurogênese, Instituto de Biofísica Carlos Chagas Filho, Universidade Federal do Rio de Janeiro, Rio de Janeiro 21941-902, Brazil; phvictorino.phv@gmail.com (P.H.V.); milamarra@hotmail.com (C.M.); rlinden@biof.ufrj.br (R.L.); 2Personalized Genomics Laboratory, Center for Computational Systems Biology, Prairie View A&M University, Prairie View, TX 77446, USA; daiacobas@pvamu.edu; 3Dominick P. Purpura Department of Neuroscience, Albert Einstein College of Medicine, Bronx, NY 10461, USA; david.spray@einsteinmed.org; 4Department of Pathology, New York Medical College, Valhalla, NY 10595, USA; sandaiacobas@gmail.com; 5Laboratório de Biologia Estrutural, Instituto Oswaldo Cruz, Fiocruz, Rio de Janeiro 21040-360, Brazil

**Keywords:** glaucoma, retinal ganglion cell degeneration, microarray, genes coordination, Notch signaling pathway, complement cascade

## Abstract

Glaucoma is a multifactorial neurodegenerative disease, characterized by degeneration of the retinal ganglion cells (RGCs). There has been little progress in developing efficient strategies for neuroprotection in glaucoma. We profiled the retina transcriptome of Lister Hooded rats at 2 weeks after optic nerve crush (ONC) and analyzed the data from the genomic fabric paradigm (GFP) to bring additional insights into the molecular mechanisms of the retinal remodeling after induction of RGC degeneration. GFP considers three independent characteristics for the expression of each gene: level, variability, and correlation with each other gene. Thus, the 17,657 quantified genes in our study generated a total of 155,911,310 values to analyze. This represents 8830x more data per condition than a traditional transcriptomic analysis. ONC led to a 57% reduction in RGC numbers as detected by retrograde labeling with 1,1′-dioctadecyl-3,3,3,3′-tetramethylindocarbocyanine perchlorate (DiI). We observed a higher relative expression variability after ONC. Gene expression stability was used as a measure of transcription control and disclosed a robust reduction in the number of very stably expressed genes. Predicted protein–protein interaction (PPI) analysis with STRING revealed axon and neuron projection as mostly decreased processes, consistent with RGC degeneration. Conversely, immune response PPIs were found among upregulated genes. Enrichment analysis showed that complement cascade and Notch signaling pathway, as well as oxidative stress and kit receptor pathway were affected after ONC. To expand our studies of altered molecular pathways, we examined the pairwise coordination of gene expressions within each pathway and within the entire transcriptome using Pearson correlations. ONC increased the number of synergistically coordinated pairs of genes and the number of similar profiles mainly in complement cascade and Notch signaling pathway. This deep bioinformatic study provided novel insights beyond the regulation of individual gene expression and disclosed changes in the control of expression of complement cascade and Notch signaling functional pathways that may be relevant for both RGC degeneration and remodeling of the retinal tissue after ONC.

## 1. Introduction

Glaucoma is a multifactorial disease characterized by degeneration of the neurons known as retinal ganglion cells (RGCs) and their long axons which transmit visual information from the retina to the brain [1]. A variety of in vivo animal models have been used to unravel the mechanisms of both RGC degeneration and reorganization of the retina. The animals were subjected to either acute or chronic elevation of intraocular pressure, transection, or crush of the optic nerve, as well as the spontaneous glaucoma-like disease (on mice of the DBA/2J strain) [2,3,4]. However, there has been little progress in developing efficient strategies for neuroprotection in glaucoma. 

The retina is a complex, multilayered tissue of the central nervous system (CNS) composed of diverse neurons, glia, and vascular cells, as well as a rich extracellular matrix. Recent studies of single-cell RNA-sequencing disclosed cell type-specific molecular markers [5]. Laser capture microdissection was also used to isolate individual cells from the retinal tissue and compare genes expressed in RGCs of either normal or glaucomatous rat retina [6]. However, similar to other neurodegenerative conditions [7], mechanisms involved in glaucomatous neurodegeneration appear to depend on a complex interplay of signaling pathways and often involve multicellular networks [8,9,10]. The systemic character of such events suggests that an examination of the whole tissue transcriptional landscape may help guide both further investigations into the pathophysiology, as well as the development of novel therapeutic approaches to glaucoma.

Transcriptional changes following either acute or chronic optic nerve injury have been identified in animal models ranging from zebrafish to non-human primates [11,12,13]. It is usually assumed that insights into functional changes caused by RGC injury in rodents may help elucidate the course of glaucomatous neurodegeneration in humans. Microarray-based transcriptome analyses are currently available for rodent models of glaucoma [6,14,15], retinal and optic disc injury [16,17], and ischemic damage [18].

Overall, the data suggest evolving stages associated with altered gene expression, roughly defined as follows: an acute phase, within hours, with transient upregulation of both immediate/early response genes and inflammatory responses [10,19,20]; a sub-acute phase, between 1 and 3 days, characterized by the expression of cell-cycle and cell death genes [21]; and a late chronic phase, at 5 to 7 days, characterized by the expression of genes involved in structural remodeling of neurons and glia [15,16,22,23,24,25]. In the earliest stage, genes related with triggering of apoptosis have been found to be upregulated in response to retinal injury [22,26,27]. In the sub-acute phase, in line with apoptotic RGC death following optic nerve damage, studies identified upregulation of genes related with the execution phase of apoptosis, as well as downregulation of cytoprotective and anti-apoptotic genes [28]. The upregulated pro-apoptotic genes include caspase 3, *tumor necrosis factor receptor type 1 associated death domain* (*TRADD*), *tumor necrosis factor receptor superfamily member 1a* (*TNFR1a*), and *BCL2 associated X apoptosis regulator* (*Bax*). Downregulated cytoprotective and anti-apoptotic genes include *X-linked inhibitor of apoptosis* (*XIAP*), *mitogen activated protein kinase 1* (*Mapk1*), *Ras/Rac guanine nucleotide exchange factor 1* (*Sos1*), *c-Raf-1*, and *YY1* [29]. In later stages, only elevated intraocular pressure and optic nerve transection models have been explored by microarray analysis, and retinal changes were found to be associated with altered expression of genes with roles in regeneration, synaptic plasticity, axonogenesis, neuronal projections, and neuron differentiation [17,20,30,31,32].

In the present study, we used Agilent gene expression microarrays and deep computational approaches to profile the expression patterns of differentially expressed genes in the rat retina at 14 days after optic nerve crush (ONC) compared to control (CTR) sham operated animals. ONC is an acute, mechanical injury to the nerve that leads to gradual RGC apoptosis and has been widely utilized to examine glaucomatous disease pathophysiology [33]. At 14 days after the lesion, a small number of RGCs remain in the retina, and thus the modified transcriptome is likely to reflect mainly the adaptation of the tissue to the loss of those [34]. 

Enrichment analysis performed in accordance with previous studies [35,36] shows complement cascade and Notch signaling pathway components as major players in the retina 14 days after ONC and, to a lesser extent, also oxidative stress and kit receptor signaling pathways were also altered. We went beyond the enrichment analysis to extract more information of the transcriptome such as the relative expression variability (REV) and the derived gene expression stability (GES). Moreover, we analyzed the coordination of gene expressions within each enriched pathway and with the entire transcriptome to determine the ONC-induced remodeling of the associated genomic fabrics. Thus, for the 17,657 quantified genes, our study generated a total of 155,911,310 values to analyze, that is, 8830x more data per condition than a traditional transcriptomic analysis. The genomic fabric of a given pathway was defined as the transcriptome associated to the most interconnected and stably expressed gene network responsible for that pathway [37]. These deep bioinformatic analyses provide additional insight into the predicted mechanisms for both RGC death and adaptation of the retinal tissue following ONC.

## 2. Materials and Methods

### 2.1. Animal Handling 

Experiments were performed on 8-week-old female Lister Hooded rats (*n* = 43), housed in plastic cages, on a 12-h light–dark cycle, with water and food ad libitum. All experiments were carried out in accord with the ARVO Statement for the Use of Animals in Ophthalmic and Vision Research, and the protocol approved by the Institutional Animal Experimentation Ethics Committee (UFRJ#01200.001568/2013-87).

### 2.2. Optic Nerve Crush

The rats were divided into 2 experimental groups: optic nerve crush (*n* = 24, hereafter denoted by ONC) and control, sham-operated (*n* = 19, denoted by CTR). Rats were deeply anesthetized by intraperitoneal injection of a mixture of xylazine (10 mg/kg body weight) and ketamine (60 mg/kg body weight). Under a stereoscopic microscope, the right optic nerve was accessed through an incision made in the superior orbital rim, and the conjunctiva was dissected with forceps towards the back of the eye to expose the retrobulbar portion of the optic nerve. The lesion was made by crushing the optic nerve for 10 s at 3 mm from the optic disc, using a pair of watchmaker’s forceps. Sham operated rats underwent the same procedure, except that the forceps were not closed. Animals were euthanized 2 weeks after the procedure.

### 2.3. Quantification of Retinal Ganglion Cell Survival

Cell survival was quantified on the basis of retrograde labeling with the lipophilic tracer DiI (1,1′-dioctadecyl-3,3,3,3′-tetramethylindocarbocyanine perchlorate, Invitrogen) in 11 CTR and 16 ONC retinas. DiI was bilaterally injected into the superior colliculus of 1-week-old neonate rats, which led to the labeling of virtually all the RGCs [38]. Seven weeks after DiI injection, rats underwent unilateral ONC. Two weeks after ONC, animals were euthanized by inhalation of carbon dioxide. At this time point, the cells that remained alive were detectable by DiI labeling and quantified. Upon dissection of the eyeballs, retinas in phosphate-buffered saline (PBS; Sigma) were dissected and flattened as whole-mounts by making 4 radial cuts, followed by fixation with 4% paraformaldehyde for 15 min at room temperature, 3 rounds of washing with PBS, and counterstaining with Sytox Green (Thermo Fisher Scientific, Waltham, Massachusetts, USA) to visualize cell nuclei. Whole retinas were mounted vitreal side-up on subbed slides, covered with anti-fading mounting media, and examined in a confocal epifluorescence microscope (LSM 510 Meta, Zeiss) using a Plan-Neofluar 40x/1.3 objective. Eight photos were taken from each quadrant of the retina, of which 2 were from central retina (≈0.9 mm from optic disc), 3 from mid-retina (≈2.0 mm from optic disc), and 3 from peripheral retina (≈3.7 mm from optic disc), to a total of 32 photos per retina. DiI+ RGCs were manually counted in a double-blind manner based on morphology to exclude microglial cells with translocated DiI and averaged across evenly distributed fields of each retina. Statistical analysis was performed through an unpaired two-tailed *t*-test using the software GraphPad Prism 9.0.1 (1992–2021 GraphPad Software, Inc.).

### 2.4. Microarray Gene Expression

Animals were euthanized by inhalation of carbon dioxide. Each retina was freshly dissected with clean instruments and immediately frozen in liquid nitrogen. For each experimental group (CTR or ONC), 4 separated retinas were profiled individually. We applied an optimized protocol [39] for RNA extraction (Qiagen RNeasy mini-kit; Qiagen, Germantown, MD, USA), reverse transcription (adding fluorescent tags), and hybridization onto Agilent G2519F 60mer two-color gene expression rat 4×44k arrays (Agilent, Santa Clara, CA, USA) in the “multiple yellow” design that provides maximum flexibility in comparing the conditions and 100% usage of the resources [40]. RNA concentrations before and after reverse transcription were estimated with a NanoDrop ND2000 Spectrophotometer, and purity was assessed with Agilent RNA 6000 Nano kit in an Agilent 2100 Bioanalyzer (Santa Clara, CA, USA). RIN (RNA Integrity Number) of at least 8.0 was considered as satisfactory. RINs of the selected ONC samples were 8.20, 8.40, 9.10, and 8.50, while those for the CTR samples were 8.40, 8.20, 8.50, and 8.60. In each array, 825 ng of Cy3 (green, g)- or Cy5 (red, r)-labeled RNA from each retina were co-hybridized for 17 h at 65 °C with that of the same experimental group in the combinations: CTR1(g)CTR2(r), CTR3(g)CTR4(r), ONC1(g)ONC2(r), and ONC3(g)ONC4(r). Chips were scanned with an Agilent G2539A dual laser scanner at 5 μm pixel size/20-bit, and raw data were collected with Agilent Feature Extraction software v. 11.1.1.

### 2.5. Data Processing

We disregarded all spots showing signs of local corruption or with foreground fluorescence less than twice their background in any one of the 8 profiled samples. Data were normalized by an in-house-developed, iterative method that alternates intra- and inter-array normalization to the median of the background-subtracted fluorescence of the valid spots until the overall maximum error of estimate reached less than 5% [41]. Normalized expression levels were organized into redundancy groups composed of all spots probing the same gene and represented by the weighted average of the values of individual spots.

The REV (REV = median of the Bonferroni like-corrected chi-squared interval estimate of the pooled coefficient of variation) was taken as a statistical estimate of the expression variability of one gene among biological replicas [42].
(1)$$REV_{i}^{\left(condition\right)}=\underset{\mathrm{correction}\;\mathrm{coefficient}}{\underset{︸}{\frac{1}{2}\left(\sqrt{\frac{r_{i}}{\chi^{2}\left(r_{i};0.975\right)}}+\sqrt{\frac{r_{i}}{\chi^{2}\left(r_{i};0.025\right)}}\right)}}\sqrt{\underset{\mathrm{pooled}\;\mathrm{CV}}{\underset{︸}{\frac{1}{R_{i}}\sum_{k=1}^{R_{i}}{\left(\frac{s_{ik}^{\left(condition\right)}}{\mu_{ik}^{\left(condition\right)}}\right)}^{2}}}}\times 100\%$$
where:condition = ONC, CTR*µ_ik_ =* average expression level of gene *I* probed by spot *k* (=1, …, *R_i_*) in the 4 biological replicas*s_ik_ =* standard deviation of the expression level of gene *I* probed by spot *k*.*r_i_ =* 4*R_i_* − 1 = number of degrees of freedom*R_i_ =* number of microarray spots probing redundantly gene *i*

The genes were then ordered according to decreasing variability, such that the first percentile (GES < 1) contained the most unstably expressed and the 100th percentile (GES > 99) the most stably expressed genes [43].

A gene was scored as significantly regulated in ONC with respect to CTR if the absolute fold-change |x| exceeded the cut-off (CUT) computed for that gene (Equation (1)), and the *p* value of the heteroscedastic *t*-test for the equal expressions was < 0.05. Thus, we replaced an arbitrary uniform absolute fold-change cut-off (e. g. 1.5×) for all genes with a value that accounts for the combined contributions of the technical noise and biological expression variability of each gene independently [44]. This composite criterion, in which the cut-offs were computed with Bonferroni-type corrections applied to the redundancy groups [45,46], eliminated most of the false positives without increasing the number of false negatives (Equation (2)).
(2)$$\begin{array}{l} \left|x_{i}^{\left(CTR\to ONC\right)}\right|>\left|CUT_{i}^{\left(CTR\to ONC\right)}\right|=1+\frac{1}{100}\sqrt{2\left({\left(REV_{i}^{\left(CTR\right)}\right)}^{2}+{\left(REV_{i}^{\left(ONC\right)}\right)}^{2}\right)}\;,\;where:\\ x_{i}^{\left(CTR\to ONC\right)}=\left\{\left.\begin{array}{cc} \frac{\mu_{i}^{\left(ONC\right)}}{\mu_{i}^{\left(CTR\right)}} & ,\;\mathrm{if}\;\mu_{i}^{\left(ONC\right)}\geq \mu_{i}^{\left(CTR\right)}\\ -\frac{\mu_{i}^{\left(CTR\right)}}{\mu_{i}^{\left(ONC\right)}} & ,\;\mathrm{if}\;\mu_{i}^{\left(ONC\right)}<\mu_{i}^{\left(CTR\right)} \end{array}\right|\right.\;\mu_{i}^{\left(ONC/CTR\right)}=\frac{1}{R_{i}}\sum_{k=1}^{R_{i}}\mu_{ik}^{\left(ONC/CTR\right)} \end{array}$$
|*a*|= absolut value of *a*

### 2.6. Analysis of Predicted Protein–Protein Interactions

Interactions among proteins encoded by differentially expressed genes were assessed using STRING (https://string-db.org/, [47,48]), which provides uniquely comprehensive coverage and ease of access to information of both experimental and predicted interaction. A protein–protein interaction network was constructed, in which the interactions of ONC vs. CTR differentially expressed genes (DEGs) were mapped to STRING on the basis of information including the sequence characters and structures. Each protein–protein interaction stored in the STRING database receives a confidence score between zero and one. These scores indicate the estimated probability that an interaction is biologically significant, specific, and reproducible. To highlight the most biologically relevant interactions, we used a confidence score cut-off of 0.4.

### 2.7. Enrichment Analyses 

PathVisio3 (www.pathvisio.org, [49]) was used to identify pathways significantly altered, and the Gene Ontology Knowledgebase (www.geneontology.org) was used to identify main biological processes, molecular functions, and cellular components affected by ONC. The pathways were ranked on the basis of a standardized difference indicator (Z score). Pathways with Z score > 2 and *p*-value < 0.05 were accepted as significantly altered. Graphical representation of the molecular pathways was based on the Kyoto Encyclopedia for Genes and Genomes (KEGG; www.genome.jp/kegg/, Kanehisa Laboratories, Japan [50]). 

The EnrichR platform (http://amp.pharm.mssm.edu/Enrichr/ [51]) contains a collection of diverse gene set libraries available for analysis and it was used to disclose disease-related perturbations by differentially expressed genes, according to the Gene Expression Omnibus (GEO) database.

Panther (Protein ANalysis THrough Evolutionary Relationships) classification system (http://www.pantherdb.org/) version 15.0 was used to analyze enrichment of genes within a specific subset of genes affected by ONC. 

### 2.8. Coordination of Expression

We used the variation in gene expression among multiple biological replicas to calculate the pair-wise Pearson correlation coefficients p between the levels of expression of gene pairs. Two genes were scored as synergistically expressed if their expression levels had a positive covariance within biological replicas, antagonistically expressed when they manifested opposite tendencies (i.e., negative covariance), or independently expressed when their transcription levels were not correlated (close to zero covariance). In the case of 4 biological replicas, the (*p* < 0.05) significant synergistically expressed genes had ρ > 0.90, antagonistically expressed genes had ρ < −0.90, and independently expressed genes had |ρ| < 0.05. The set of correlation coefficients between the expression level of a particular gene and of each other gene within the biological replicas forms the coordination profile of that gene. The profiles of two genes can be similar, opposite, or neutral. Two genes were considered as having similar coordination profiles when there was significant overlap of their synergistically, antagonistically, and independently expressed partners; genes were considered as having opposite coordination profiles when most of the synergistically expressed partners of one gene were antagonistically expressed partners for the other, and most of the independently expressed partners of one gene were synergistically or antagonistically expressed for the other [45,46].

### 2.9. Quantitative Real-Time PCR

To validate selected genes, we used a new set of animals (*n* = 4 per group) for qRT-PCR. Total RNA (2 μg) isolated from retinas using TRIzol reagent (Invitrogen, Carlsbad, CA, USA) was reversely transcribed into complementary DNA (cDNA) with the SuperScript III Reverse Transcriptase (Invitrogen, Carlsbad, CA, USA). qRT-PCR was performed with SYBR Green Mix (Promega, Madison, WI, USA) in a 7500 Real-Time PCR system (Applied Biosystems, Waltham, MA, UK) in order to detect the expression of *Cd74* (Cd74 molecule, major histocompatibility complex, class II invariant chain), C3 (complement component 3), *Tubb3* (beta-III-tubulin), *Nefm* (neurofilament medium), *Nell2* (neural EGFL-like 2), *Cyba* (cytochrome b-245 light chain), and *Fyn* (FYN proto-oncogene, Src family tyrosine kinase). Results were normalized to the expression of 2 housekeeping genes: rat glyceraldehyde-3-phosphate dehydrogenase (GAPDH) and mitogen-activated protein kinase 1 (MAPK1). Primers were *Cd74* forward 5’- GAACCTGCAACTGGAGAACC-3’; reverse 5’- CTTCGTAAGCAGGTGCATCA-3’; *C3* forward 5’- GCATCAGTCACAGGATCAGGTCA-3’; reverse 5’- ATCAAAATCATCCGACAGCTCTATC-3’; *Tubb3* forward 5’- TATGTGCCCAGAGCCATTC3’; reverse 5’- CACCACTCTGACCHAAGATAAA-3’; *Nefm* forward 5’- GCTGCAGTCCAAGAGCATTG-3’; reverse 5’- CTGGATGGTGTCCTGGTAGC-3’; *Nell2* forward 5’- GGCTTTAGATTGCCCCGAGT-3’; reverse 5’- CGTTCAGGTTCCTGCAGACT-3’; *Cyba* forward 5’- GTGAGCAGTGGACTCCCATT-3’; reverse 5’- GTAGGTGGCTGCTTGATGGT-3’; *Fyn* forward 5’- GACCATGTGAATGTGCTCCG-3’; reverse 5’- ACTGACCTTTTGCCACGACT-3’; *GAPDH* forward 5’- GACATGCCGCCTGGAGAAAC-3’; reverse 5’- AGCCCAGGATGCCCTTTAGT-3’; *MAPK1* forward 5’-TGTTGCAGATCCAGACCATG-3’; reverse 5’-CAGCCCACAGACCAAATATCA-3’.

The PCR program was as follows: denaturation at 95 °C for 5 s, annealing at 60 °C for 30 s, and elongation at 68 °C for 20 s for 40 cycles. Each sample was applied in technical duplicates, and fold changes were calculated using the 2-^(ΔΔCT)^ method of relative quantification. For each gene tested, all experimental and CTR retinas were assessed in the same experiment. The data were normalized to the geometric mean of the 2 housekeeper genes and expressed as fold change. Statistical analysis was performed using GraphPad Prism Software (mean ± standard error of the mean, SEM) using unpaired two-tailed Student’s *t*-test. Differences were accepted as significant when *p* < 0.05.

## 3. Results

### 3.1. Histopathological Observations

At 14 days after ONC, there was a robust loss of RGCs retrogradely labeled with DiI, as expected. CTR retinas displayed an average of 1.780 ± 134 cells/mm^2^, whereas ONC retinas showed a reduction of approximately 60% of RGCs, to 768 ± 65 cells/mm^2^ (Figure 1A–C).

### 3.2. Transcriptomic Alterations in Rat Retina after ONC

The experimental details, as well as both raw and normalized expression data from the microarrays, were deposited and are publicly available at Gene Expression Omnibus (GEO; accession number GSE133563). Among 17,657 unigenes quantified in our microarrays, expression of 193 (1.1%) was significantly altered in ONC samples with respect to CTR, of which 127 (65.8%) were up- and 66 (34.2%) were downregulated. Table 1 presents the most downregulated 50 genes and Table 2 presents the most upregulated 50 genes. 

Quantitative RT-PCR was used to validate two of the highest upregulated genes (*Cd74* and *C3*) and three markers of RGCs (*Tubb3*, *Nefm*, *Nell2*) that were found to be downregulated in ONC retinas as compared to CTR (Table 3). Cd74 and C3 were found to be upregulated by qPCR by 28.56- and 3.24-fold, respectively. *Tubb3*, *Nefm*, and *Nell2* were downregulated by −3.93-, −19.27-, and −4.15-fold in ONC retinas when compared to CTR retinas, consistent with the results of the arrays (Table 3). 

As expected, many of the downregulated genes found in our arrays have been previously described as markers of RGCs [5,6] (Figure 1D,E and Table 4). Notably, 13 RGC markers had the highest downregulation fold changes, as for example, *peripherin* (−12.7x); *neurofilament heavy*, *medium*, and *light polypeptides* (−11.4x, −11.8x, and −6.8x, respectively); *gamma synuclein* (−10.8x); *POU class 4 homeobox 2* (or *Brn3A*, −7.3x); Thy-1 cell surface antigen (−3.9x); *synaptotagmin II* (-2.4x); *tubulin*; *beta 3 class III* (-1.9x); *internexin neuronal intermediate filament protein* (−1.9x); *alpha-internexin* (−1.9x); and *pannexin 2* (−1.89x) ( Table 1; Table 3 and the green points in Figure 1E). In addition, the RGC neurotrophic factor *neuritin 1* [52] was also strongly downregulated after ONC, with a −18.8x decrease when compared to CTRs. The EnrichR software, a tool to determine disease-enriched perturbations in the transcriptome that uses the GEO database, showed that among the downregulated genes at 14 days after ONC, the top disease signature was “glaucoma associated with systemic syndromes” (Appendix A). These results further validate the characterization of RGC degeneration promoted by ONC.

### 3.3. Protein–Protein Interaction Encoded by the Most Downregulated Genes

We used the STRING platform to predict protein–protein interaction (PPI) networks. Among the downregulated genes, only 22 showed a network of 21 interactions, with a medium confidence score of 0.4. Enrichment analysis using the Gene Ontology database indicated that the main biological processes altered by ONC were intermediate filament cytoskeleton organization, intermediate filament bundle assembly, and response to acrylamide, all of which include neurofilaments (Figure 2A), a major component of the optic nerve axons. Among molecular functions, ONC affected networks of protein binding, binding, and protein heterodimerization activity (Figure 2B). Finally, among cellular components, ONC affected cell projection, axon projection, and neuron projection (Figure 2C). All such functional interactions are consistent with retrograde degeneration as a consequence of ONC.

### 3.4. Protein–Protein Interaction Encoded by The Most Upregulated Genes

Consistent with previous studies [6,15,16,17,25], the group of 50 most upregulated genes (Table 2) includes inflammatory response-related genes, such as Cd74 molecule, major complex, class II invariant chain (25.6x), H-2 class II histocompatibility antigen gamma chain (MHC class II-associated invariant chain) (Ia antigen-associated invariant chain) (Ii) (CD74 antigen) (11.6x), follistatin-like 3 (5.3x), serpin peptidase inhibitor, clade G, member 1 (3.5x), and Eph receptor A2 (3.3x). Similar to the downregulated set of genes, the EnrichR platform indicated glaucomatous/neurodegenerative transcriptomic profile as the main systemic syndrome or disease (Appendix A). Further, PPI analysis showed that out of the upregulated genes, 64 had at least one interaction that formed a network of 129 interactions with a medium confidence score of 0.4 (Figure 3). Followed by enrichment analysis, the main biological processes identified were immune system, regulation of cell proliferation, and innate immune response (Figure 3A). Among molecular functions, binding, protein binding, and ion binding categories presented the most pronounced interactions, with 33, 25, and 24 altered genes, respectively (Figure 3B). As for cellular components, interactions were found in extracellular space, extracellular region, and extracellular region part categories (Figure 3C). Both molecular function and cellular components are consistent with immune system signaling, the main biological process identified among the upregulated genes.

### 3.5. ONC Altered The Stability of Gene Expression

In addition to the average expression level, the REV among biological replicates provided an important parameter of comparison between CTR and ONC retinas. We found that the REV profile of genes after ONC was very similar to CTR retinas, but with a slight shift towards higher variability (Appendix A). REV values were then used to estimate the GES of individual genes in each condition. We found a decrease in the number of very stably expressed genes (REV < 10), from 232 in CTR retinas to 48 after ONC. In order to analyze whether the GES values of individual genes were similar between the experimental groups, the GES value for each spot from CTR retinas was plotted against the GES value for the same spot from ONC retinas (Appendix A). Only a few genes, delimited by the red dotted circles in Appendix A, displayed clearly distinct GES values between the two conditions. Panther was used to analyze enrichment of the genes within the red dotted circles (Appendix A), indicating that most of these genes are related to eye structure and development (Table 5). This result is in agreement with the structural changes in the retina after ONC as expressed by RGC degeneration (Figure 1). In particular, *crystallin* genes underwent a dramatic change in their GES, from stable expression in CTR sample (GES > 60) to a very unstable expression (GES < 6), indicating active participation in the retinal changes at 14 days after ONC.

### 3.6. Pathway Enrichment Analysis in ONC

We used the PathVisio3 platform to identify which biological pathways were the most affected among genes upregulated by ONC. “Complement activation, classical pathway” and “complement and coagulation cascades” pathways appeared with the highest Z scores, followed by “delta-Notch signaling pathway” and “Notch signaling pathways”, all related with immune response (Table 6). Additionally, “oxidative stress” and “kit receptor signaling pathway” were found significantly altered in ONC retinas when compared with controls. In addition, many among the top 50 upregulated genes belonged to the complement cascade and Notch signaling pathways (Table 6). One representant from “complement cascade” (*C3*), “oxidative stress” (*Cyba*), and “kit receptor signaling” (*Fyn*) pathways were further validated by RT-qPCR (Table 3). We plotted the results obtained with PathVisio3 into pathway templates from either the Kyoto Encyclopedia of Genes and Genomes (KEGG) or WikiPathway to visualize significantly altered genes, both up- and downregulated, in all pathways. We found more significantly upregulated genes (red), with higher fold-change within the “complement cascade pathway”, “Notch-” and “delta-Notch” signaling pathways (Figure 4 and Figure 5, Appendix A) than within oxidative stress and kit receptor signaling pathway (Appendix A). 

In the “complement cascade pathway”, we found no downregulated genes and five significantly upregulated genes (red) belonging to both “classical” and to the “alternative pathways”: *C3* (2.17x), *Cfb* (2.26x), *C1s* (2.43x), *C4* (2.41x), and *Serping1* (3.49x) (Figure 4). Two other genes belonging to the “membrane attack complex” were also upregulated: *vitronectin* (1.83x) and *clusterin* (1.71x) (Figure 4A,B). Next, we examined the number of coordination interactions (synergistic and antagonistic) between gene pairs with significant pairwise Pearson correlation coefficients from the “complement cascade pathway”. This analysis allows an estimate of the degree of network interlinkage within the pathway. Expression coordination of two genes may be an indication that the encoded proteins are linked in one or more functional pathways, and therefore their expression levels should be in a particular proportion [53,54]. Among the CTR retinas, we found 29 positive coordinations (“synergisms”), represented by magenta lines in Figure 4A. After the ONC, the number of synergisms raised to 36, and two independent coordinations were found (black dotted lines, Figure 4B). For all pathways examined in PathVisio3, no negative coordination (“antagonism”) was found, while an increase in positive coordinations was observed following ONC.

We also examined the coordination profiles of all genes from the “complement cascade pathway” against all other genes in our microarray datasets in each experimental condition. Such analysis identifies gene pairs with either similar or opposite coordination profiles on the basis of the overlap of their synergistically, antagonistically, or independently expressed partners. We found that the synergistic coordination among the “complement cascade pathway” genes increased from 21% in CTR retina to 36% after ONC, indicating a stronger alignment of the genes to the network, whereas there were no opposite coordination profiles. As an example, the coordination profiles of the pairs *C3*-*C1qb*, *Serping1*-*C1qb*, and *Serping1*-*Vtn* are shown in Figure 4C–E. In CTR retinas, the coordination profiles of the paired genes were neutral (black dots), whereas ONC retinas showed a robust increase in similarity (oranges dots), with *R^2^* values of at least 0.9. 

The “delta-Notch” and “Notch signaling pathways” were also significantly altered by ONC, with five genes upregulated when compared to CTR retinas: *Cntf* (1.55x), *Notch1* (1.94x), *Fhl1* (1.53x), *Smad1* (2.12x), and *Dtx2* (1.51x) (Table 6). The WikiPathway template was used to visualize the altered expression in Notch1 and Deltex2 genes (Figure 5A,B), and the number of coordination interactions between pairs of genes by Pearson’s correlations is represented by magenta (synergism) or black dotted (independent coordination) arrows. The number of synergistically expressed gene pairs increased from 20 in the CTR condition to 52 after ONC (Figure 5A,B). No independently expressed genes were observed in the ONC dataset. To further assess whether genes in the “Notch signaling pathway” are committed to this network, we examined the coordination profiles of genes from this pathway against all other genes in our microarray datasets in each experimental condition. We found an increase of similar coordination from 22% in CTR retinas to 28% after ONC. As an example, two neutral profiles (black dots) in the CTR retinas for the *Dll1*-*Notch1*, *Dvl3*-*Dtx3*, and *Notch4*-*Notch1* pairs (*R*^2^ = 0.08, 0.18, and 0.63, respectively), changed to similar coordination (orange dots) (*R*^2^ > 0.9) after ONC (Figure 5C–E). These results indicate that ONC induced strong gene networking in the “Notch signaling pathway”. We performed similar analysis of Pearson’s correlations for genes present in the delta-Notch signaling pathway (Appendix A). Using KEGG templates, we showed that ONC led to upregulation of four transcripts (*Notch1*, *Cntf*, *Smad1*, and *Fhl1*) and no significant downregulated genes (Appendix A). Whereas the CTR group had 38 synergistic correlations, ONC led to an increase in the number of such interactions to 76 (Appendix A). 

As previously stated, another two pathways were altered by ONC with significant Z scores, as indicated by PathVisio analyses: “oxidative stress” and “kit receptor pathway”. Two genes were significantly upregulated in the “oxidative stress pathway” (*Nfix* and *Cyba*), and no downregulated genes were altered by ONC (Appendix A). In the CTR retinas, *Nfix* showed synergistic interactions with *Xdh* and *Maoa*, which were abrogated in the ONC, where *Nfix* had synergism only with *Cyba* (Appendix A and Table 3). The overall number of synergisms in this pathway was increased from 57 in the CTR to 82 in the ONC groups. 

Finally, we analyzed the kit receptor signaling pathway (Appendix A). *Cblb*, *Fyn*, and *Tec* were upregulated by ONC, and no significantly downregulated genes were verified in this pathway (Appendix A and Table 3). The number of synergistic correlations was also increased by ONC in this pathway, 43 in CTR and 53 in ONC (Appendix A). We chose one representative gene from each pathway (complement, oxidative stress, and kit receptor signaling) to perform a RT-qPCR validation in a new set of retinas (Table 3) and found that *C3*, *Cyba*, and *Fyn* were consistently upregulated in rat retinas after ONC.

## 4. Discussion

In this study, we conducted a transcriptomic analysis of rat retina at 14 days after ONC, a time point when the population of RGCs was reduced to only 40% of the original population, as compared to controls. We identified differentially expressed genes, at a total of 127 genes significantly upregulated, and 66 downregulated. We examined changes in GES and found a reduction in the number of very stably expressed genes from 336 to 124. Using bioinformatic tools to propose protein interaction (STRING) and main altered molecular pathways (PathVisio3), we found that the complement cascade and Notch signaling pathway were the main affected pathways. Although protein–protein interaction determined with STRING (on the basis of data mining in genome-wide datasets) may be a good predictor of real interactions [48], we deepened our analyses with additional bioinformatics and mathematical modeling of gene networks of ONC retinas, which has been shown by our group to be a consistent method for such predictions. Coordination analysis for both pathways showed an increase in the number of synergistically expressed gene pairs and of genes with similar coordination profile.

Our analysis was carried out on whole-retina extracts, and the transcriptome profiles analyzed represent the adaptations of the retinal tissue to the RGC degeneration promoted by the ONC. Although the observed regulation is triggered by the ONC, which leads to RGC death, changes at 14 days after injury likely involve the activation of the retinal glia (i.e., astrocytes, Müller cells, and microglial cells), and also other retinal neurons besides RGCs, as signs of retinal remodeling. These changes may impact potential treatment strategies to ameliorate the secondary degeneration associated with CNS insults [34].

Unique among transcriptomic studies of the retina is that beyond the regulation of gene expression, we also disclosed REV, changes in GES, and coordination following ONC. GES score ranks the genes according to the strength of the cellular homeostatic mechanisms to keep their expression variability within narrow limits. GES, which includes the most stably expressed genes in the 100th percentile, actually identifies the genes most critical for the survival and phenotypic expression of the tissue. By contrast, the most unstably expressed genes (GES in the first percentile) most likely empower the cell to respond/adapt to the environmental fluctuations [45,46]. Moreover, the expression coordination analysis refines the functional pathways by the identification of gene pairs whose expression fluctuations among biological replicas are positively (synergistically), negatively (antagonistically), or neutrally (independently) correlated. In previous papers [43,45,46], we speculated that genes whose encoded proteins are linked in functional pathways should coordinate their expression to optimize a kind of “transcriptomic stoichiometry”. Changes in the coordination profiles indicate remodeling of the functional pathways [37,42,43,44,45,46,53,54,55]. 

Expression of individual genes depends on local conditions that, although similar, are not identical among biological replicas. We assume that expression of key genes is kept by the cellular homeostatic mechanisms within narrow intervals while that of non-key genes is less restrained to readily adapt to environmental changes. In our results, we found a higher variability in the REV profile of genes with ONC (Appendix A), as expected in pathological conditions, suggesting that CTR mechanisms are also affected.

Thus, by analyzing GES changes following the ONC, we found clues about retina cells changing priorities in controlling the expression level of certain genes. Our results identified, at 14 days after ONC, that only a few genes presented an altered profile in their GES, with enrichment in genes of eye structure (Appendix A). The *crystallin* genes were found to have relevant changes in their GES, thus acquiring a more unstable expression profile. Crystallins have chaperone-like properties involved in an increase of cellular resistance to stress-induced apoptosis [56]. Upregulation and downregulation of crystallin expression has been reported upon different cellular stresses, and those alterations are viewed as a cellular response against environmental and metabolic insults [57,58,59]. Crystallins have been related with both RGCs [60] and amacrine cells [61], supporting its active participation in the remodeling of retinal tissue after injury, possibly through a decrease in resistance to stress, as well as leading to tissue stabilization. Curiously, two other genes presented a drastic change in GES: *Adad2* (GES (CTR) = 79 to GES (ONC) = 5), a gene related with RNA editing [62], and *RGD1308160* (GES (CTR) = 61 to GES (ONC) = 10), a gene that, according to BlastP, has a loose homology with the *Caenorhabditis elegans* gene *smc-4*, involved in chromosome stabilization [63]. These data suggest that both such mechanisms are very active at 14 days after ONC.

Typical RGC-expressed genes, underrepresented in ONC retinas compared to CTR (Figure 1 D,E), reflect RGC death after optic nerve damage. Among the 66 downregulated genes, 20% have been described as expressed by ganglion cells [5,6,20]. Enrichment analysis of the downregulated genes showed cytoskeletal organization as the main biological process, with genes of intermediary filaments and of proteins that interact with them. Neurofilaments are particularly abundant in axons, and essential for their growth, maintenance of caliber, transmission of electrical impulses, velocity of impulse conduction, as well as regulation of enzyme function and the structure of linker proteins [64,65,66].

Additional biological roles have been attributed to neurofilaments such as synaptic plasticity. Marked decrease in the expression of the neurofilament genes was first shown in brain tissue from Alzheimer’s disease patients as compared to CTRs [67]. It is generally accepted that CNS disorders are divided between early and late changes, associated with dysfunction of neuronal activity caused by perturbations of synapses [68]. Neural reorganization after stroke is initiated as cellular reactions to degeneration. While neurons die in an ischemic region, axons and synapses degenerate in further brain regions, promoting regenerative responses that lead the growth of new connections among surviving neurons [69]. RGC dendritic arbors also showed significant reduction after 2 weeks of elevation of intraocular pressure, a common model of chronic glaucoma [70]. In a primate glaucoma model, a correlation was established between abnormalities of parasol RGC dendrite morphology and function [71]. Morphological changes of RGC dendritic arbors have been detected before axon thinning or soma shrinkage, suggesting that dendritic abnormalities may precede degeneration of other ganglion cell domains. Alterations of neurofilament expression in various neurological disorders not only underlie axonopathy, but also synaptic remodeling because of its vital roles in synaptic function [72,73]. Since at 14 days after ONC the retinal tissue still harbors 40% of the RGCs, the decreased expression of genes related with cytoskeletal organization, as intermediary filaments, may be representative of retinal remodeling, not restricted to the injured axon but also at the level of dendrites. This was further reinforced by the predicted protein–protein interactions (STRING) analysis that showed “axon” and “neuron projection” cellular components were significantly altered. PPI along with Pearson correlation analyses performed herein took advantage of the concept that expression of individual genes is linked to each other so that the protein amounts would respect the “stoichiometry” of the biochemical reactions [74].

Many components of the innate immune system are expressed intrinsically by retinal cells (Retinal Database, GeneNetwork.org), and innate immune networks are activated by various types of insults to the retina, such as ONC, partial transection of the optic nerve, and DBA2J mouse models of glaucoma [6,10,14,15,20,75,76]. In the present study, enrichment analysis disclosed the complement cascade as the most prominent biological process among differentially expressed genes after ONC (Appendix A), which was further confirmed by predicted PPI as “innate immune response” and “immune system process”. Complement activation was shown to be upregulated in the retina as early as 2 days after ONC [27]. Such activation, specifically including *C1q*, *C3*, and *Cfi* [19], suggests that the innate immune system plays an important role in retinal immune response through glia cells, and specifically in the response of the retina to injuries. It has been suggested that, at an early stage of a mouse glaucoma model, the complement cascade mediates synapse loss, in particular *C3* [77,78,79,80]. At late-stage glaucoma, the complement cascade is thought to play its traditional role of opsonizing and removing the cellular debris from widespread RGC death [81]. Recently, it was described that C1q marks a subset of RGC in the embryonic retina and that knockout of C3 receptor, which is only found in microglia, resulted in increased RGC numbers, indicating a direct relationship with microglia-mediated RGC elimination [82]. 

Our data showed that ONC led to an increase in synergistic coordination, most evident in the classic pathway of complement cascade, which indicated a stronger gene network in this pathway. It is particularly interesting that the *serping1* gene showed increased synergistic coordination at 14 days after ONC (Figure 4). Serping1 is a C1 inhibitor known to block the initiation of the complement cascade through the classical pathway, therefore limiting inflammation. It is indirectly involved in the production of bradykinin, a peptide that promotes inflammation by increasing the permeability of blood vessel walls and also inhibits leukocyte recruitment into ischemic cardiac tissue, an effect independent of protease inhibition [83]. Therefore, the upregulation of *serping1* in ONC retinas may help control the inflammatory response after injury, both by modulation of the complement cascade, as well as preventing infiltration of peripheral inflammatory cells through the blood–retina barrier.

Our enrichment analysis also showed upregulation of the Notch signaling pathway at 14 days after ONC. Notch signaling is known to play important roles during retinal development, in both RGC differentiation [84] and proliferation of Müller glia in acutely damaged chick retina [85]. Notch is expressed in most Müller glia cells at low levels in undamaged retina, and blockade of Notch activity prior to damage is protective to amacrine, bipolar, and ganglion cells [86]. Moreover, overexpression of Notch1 has been observed in several neurodegenerative diseases [87,88]. Besides Notch, the *deltex2* gene was also upregulated (Figure 5). A recent study showed that Dtx together with its interacting partner, Hrp48, downregulated Notch signaling and induced cell death during *Drosophila* development [89]. At 14 days after ONC, the upregulation of both *Notch* and *deltex2* suggests that Notch signaling may act upon retinal Müller cells via a non-canonical pathway, thereby positively influencing RGC degeneration. Notch signaling mediates many different intercellular communication events, influencing retinal Müller cells, ganglion cells, and microglial cells. It is known that Notch-1 signaling in neurons during ischemic conditions may enhance apoptotic signaling cascades, while activation of Notch-1 in microglial cells and leukocytes may exacerbate inflammatory processes that contribute to neuronal death [90]. Moreover, several studies found that Notch signaling serves important functions in the regulation of neurite outgrowth and maintenance. Both Sestan et al. [91] and Berezovska et al. [92] found that Notch signaling influences dendritic morphology—while activation inhibits neurite outgrowth or causes their retraction, inhibition promotes neurite extension. These data, together with our results, suggest that Notch signaling may confer a positive effect on RGC degeneration through an exacerbation of inflammatory processes as well as promoting retinal tissue remodeling by altering dendritic morphology.

Oxidative stress pathway was also found to be enriched in our analyses, and this was shown to occur due to *Cyba* and *Nfix* upregulation in ONC retinas. In the CNS, normal NAD(P)H oxidase function appears to be required for processes including neuronal signaling, but overproduction of reactive oxygen species (ROS) contributes to neurotoxicity and neurodegeneration. Conversely, RGCs were found to express NAD(P)H oxidase subunits under physiologic conditions and after ischemia [93]. NOX proteins are homologs of NAD(P)H oxidases that generate reactive oxygen species [94]. NOX proteins interact with and are stabilized by Cyba (or p22phox) in intracellular vesicles, which, when activated by cytosolic p47phox, form a protein complex that transports electrons from cytoplasmic NADPH to extracellular oxygen to generate superoxide (O2−). Cyba was also found to be increased in a rat model of diabetic retinopathy [95] and in retinal pigmented epithelial cells under angiotensin-II-induced oxidative stress [96]. 

NFI transcription factors a/b/x were shown to be temporally expressed during retinal development by retinal progenitor cells, bipolar cells, and Müller glia [97]. Moreover, *Nfix* is known to have a role on CNS development, and *Nfix*-null mice display major brain malformations in the cortex, cerebellum, and hippocampus ([98] for reviews). In the skeletal muscle, Nfix can also regulate myogenesis [99,100], however, deletion of Nfix leads to decreased muscle regeneration and has been implicated in the pathogenesis of muscular dystrophies [101]. The knockdown of *Nfix* in a myoblast cell line was protective against oxidative stress [102], thus indicating that this transcription factor may have a versatile role in health and disease. In the context of retinal degeneration, the upregulation of Cyba and Nfix transcripts reinforces the hypothesis that oxidative stress takes place and accelerates RGC death in ONC models [103]. 

Oxidative stress and inflammation are closely related pathophysiological processes, one of which can be easily induced by another. Thus, both processes are simultaneously found in many chronical pathological conditions, such as glaucoma [104]. The immune activity is a necessary intrinsic response to promote tissue cleaning, healing, and regeneration process after injury [105]. However, unbalanced or sustained immune response turns the beneficial potential into potentiation of the injury process. The mechanical injury together with hypoxia, both promoted by the ONC, could be the triggers for the activation of inflammation, oxidative stress, and metabolism alterations. Notch-1, c-Kit, and complement signaling are known as linkers between inflammation, metabolism, oxidative stress, and dendritic plasticity, acting into RGC and Müller cells [88,106,107,108,109,110]. Our results support the hypothesis that not only the events intrinsic to RGCs, but also environmental signals from other cells, such as Müller cells, are critical for cell death stimuli, and that RGC–glia interactions are critically important for different aspects of RGC neurodegeneration. Improvement of the current understanding of the role of RGC–glia interaction, mitochondria, and the immune system in neurodegeneration will potentiate the development of innovative treatment methods for neuroprotection.

## 5. Conclusions

In summary, our data indicate that ONC induces a robust, long-lasting alteration of at least two main pathways (complement cascade and Notch signaling pathway), including a significant increase in the expression and coordination of inflammation-related genes. The coordination analyses of genes within each pathway and with the entire transcriptome suggest that such biological pathways may act after ONC as major players in changes of gene expression that lead to both degeneration and remodeling of retinal tissue. Further functional studies on the effects of these genes are warranted to clarify their roles in molecular mechanisms within the retinal tissue after damage to the optic nerve.

## Figures and Tables

**Figure 1 genes-12-00403-f001:**
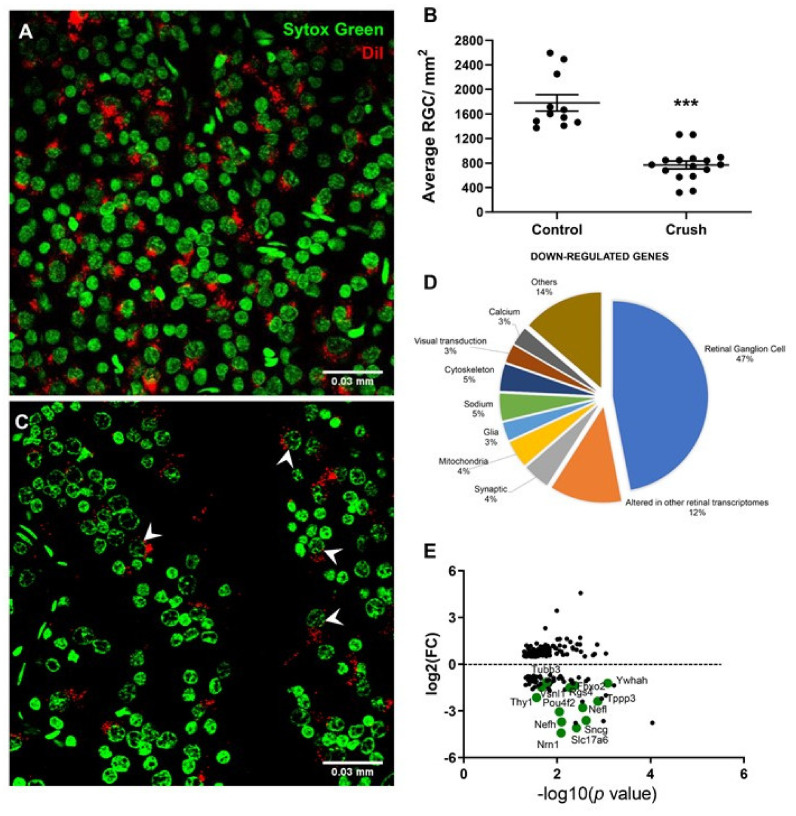
General profile of the number retinal ganglion cells (RGCs) after the optic nerve crush (ONC). (**A**) Retina from control (CTR) eye showing the pattern of RGC retrograde labeling by 1,1′-dioctadecyl-3,3,3,3′-tetramethylindocarbocyanine perchlorate (DiI; red). Nucleus of remaining RGCs (arrowheads) are shown by SYTOX Green staining in green. (**B**) Retina quantification showing a reduction of approximately 60% of DiI-retrogradely labelled RGCs. CTR retinas displayed an average of 1.780 ± 134 cells/mm^2^ and ONC retinas presented 768 ± 65 cells/mm^2^. (Difference between means (ONC *n* = 16 – CTR n = 11) -1012 ± 60.98 *p* < 0.0001.) (**C**) ONC retinas showed decreased numbers of RGC after 14 days. (**D**) Pie chart of the distribution of downregulated genes based on GeneCard and Pubmed analysis showed that the majority (47%) were expressed in RGC, and 12% were described to be decreased in other retinal microarray studies of RGC degeneration. (**E**) Differentially expressed genes after ONC were plotted as log_2_ values against their *p*-values in log_10_. RGC-specific genes that were previously shown to be altered by ONC are highlighted in green. *** *p* < 0.001, unpaired Student’s *t*-test.

**Figure 2 genes-12-00403-f002:**
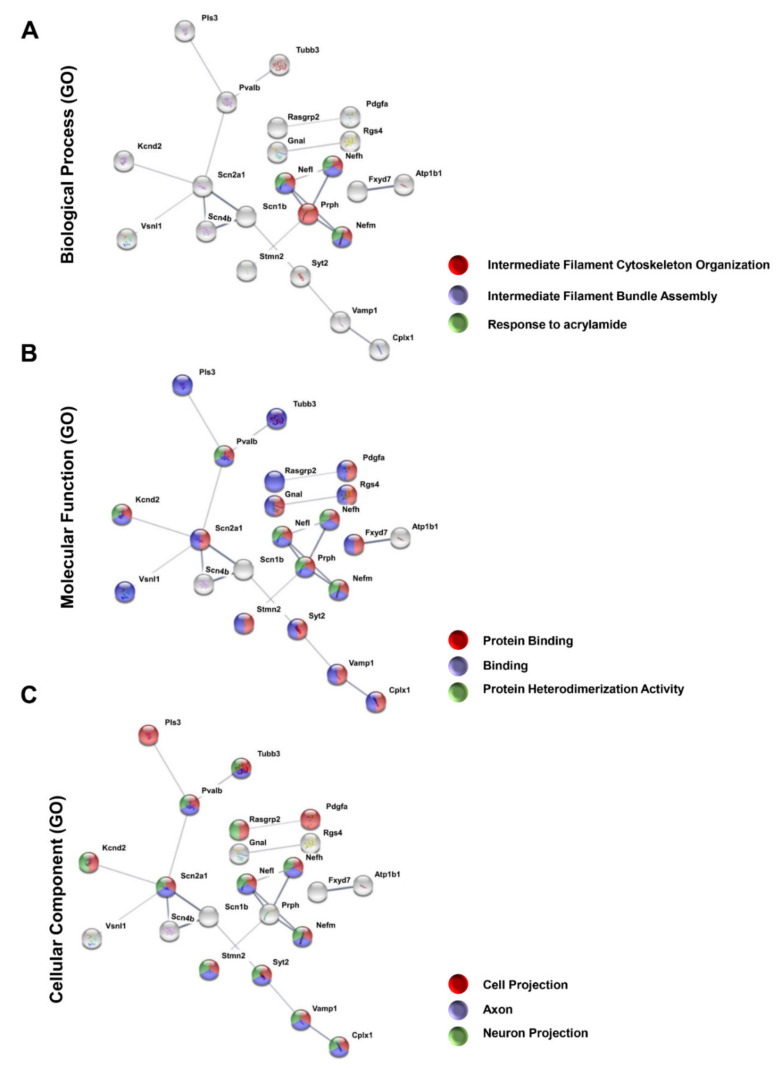
Protein–protein interaction (PPI) network composed of downregulated genes after the ONC. Among 66 downregulated genes, 21 interactions between 22 genes were found using STRING software, with a confidence cut-off of 0.4. Nodes labeled with the encoding gene symbol indicate proteins and the lines represent the corresponding interactions. The confidence score of each interaction is mapped to the line thickness (the thicker the line, the more evidence to support the interaction). The network was then enriched according to Gene Ontology database. The categories of Gene Ontology are depicted: (**A**) biological processes, (**B**) molecular function, and (**C**) cellular components, and the three most significantly enriched categories were used to color the nodes of the interaction networks.

**Figure 3 genes-12-00403-f003:**
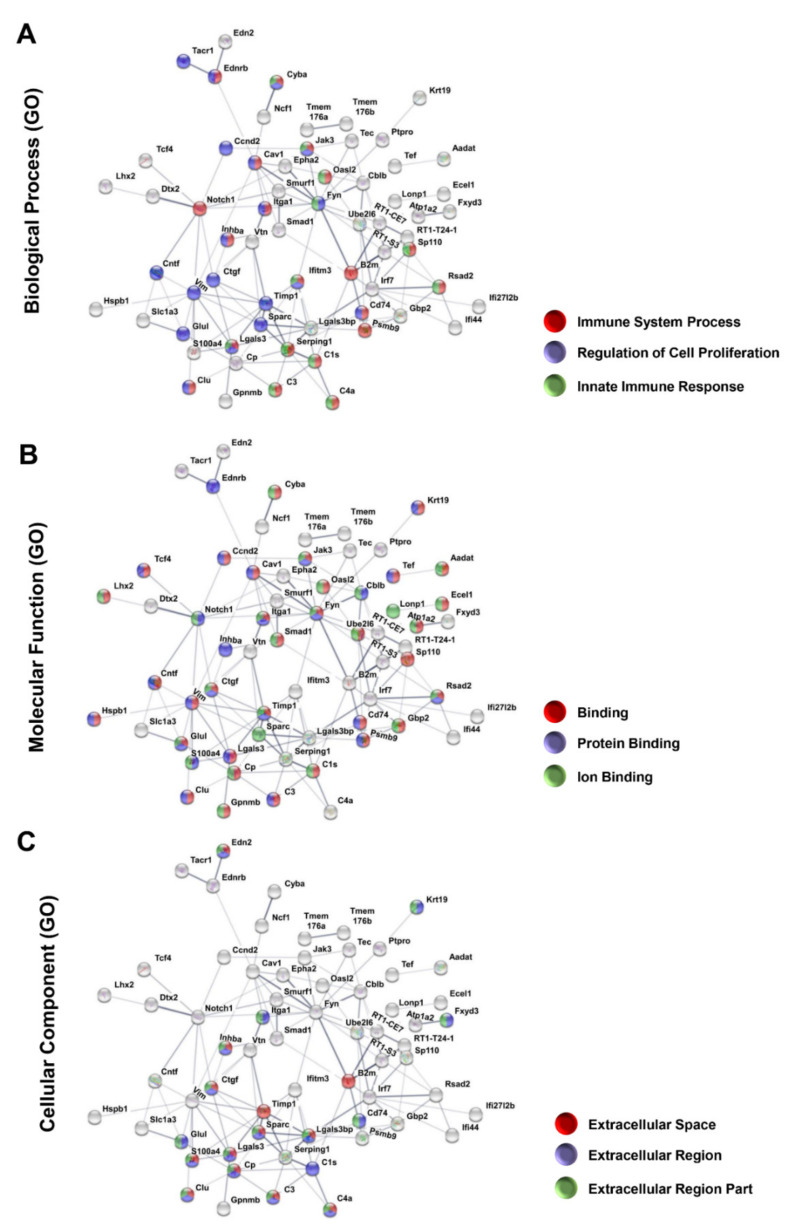
Protein–protein interaction (PPI) network composed with upregulated genes after the ONC. The 127 upregulated genes formed 129 interactions between 64 genes with a confidence cut-off of 0.4, as evidenced by STRING software. Nodes labeled with the encoding gene symbol indicate proteins and the lines represent the corresponding interactions. The confidence score of each interaction is mapped to the line thickness (the thicker the line, the more evidence to support the interaction). The network was then enriched according to the Gene Ontology database. The categories of Gene Ontology are depicted: (**A**) biological processes, (**B**) molecular function, and (**C**) cellular components, and the three most significantly enriched categories were used to color the nodes of the interaction networks.

**Figure 4 genes-12-00403-f004:**
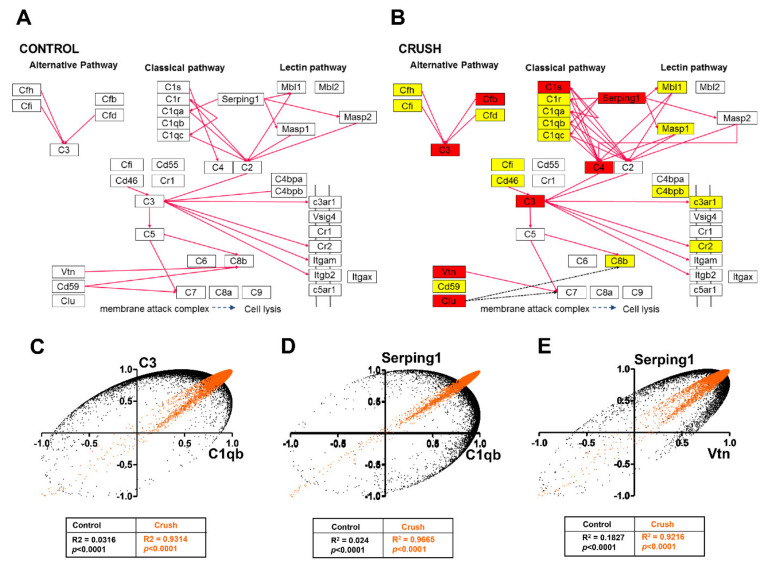
ONC affected the complement cascade pathway. The complement cascade pathway obtained from the Kyoto Encyclopedia for Genes and Genomes (KEGG) platform was used as a template to highlight the effect of the ONC in the retinas. Magenta lines in (**A**) and (**B**) represent synergistic correlations between pairs of genes and dashed black lines correspond to independent coordination. ONC induced an increase in the number of synergistic correlations. Yellow boxes in (**B**) indicate genes that showed no significant alteration in ONC versus CTR retinas, whereas red-filled boxes represent gene upregulation (> 1.5-fold change, *p* <0.05). White boxes indicate the genes that were absent in the analysis. (**C**–**E**) Plots of correlation coefficients between the expression levels of the indicated genes with each other genes differentially expressed in each experimental condition. Note a neutral coordination profile (black dots) for CTR retinas and a similar profile (orange dots) after ONC.

**Figure 5 genes-12-00403-f005:**
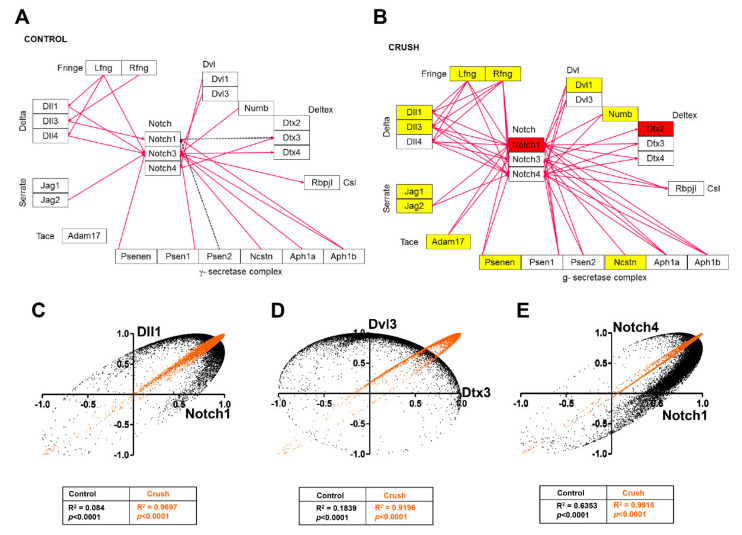
ONC affects the Notch signaling pathway. The Notch signaling pathway was obtained from the WikiPathway platform and used as a template to highlight the effect of the ONC in the retinas. Magenta lines in (**A**) and (**B**) represent synergistic correlations between pairs of genes and dashed black lines correspond to independent coordination. ONC induced an increase in the number of synergistic correlations. Yellow boxes in (**B**) indicate genes that showed no significant alteration in ONC versus CTR retinas, whereas red-filled boxes represent gene upregulation (>1.5-fold change, *p* < 0.05). White boxes indicate the genes that were absent in the analysis. (**C**–**E**) Plots of correlation coefficients between the expression levels of the indicated genes with each other genes differentially expressed in each experimental condition. Note a neutral coordination profile (black dots) for CTR retinas and a similar profile (orange dots) after ONC.

**Table 1 genes-12-00403-t001:** Top 50 downregulated genes in ONC versus CTR.

Gene Name	Gene Symbol	Fold Change	*p*-Value
Neuritin 1	*Nrn1*	−18.82	0.0080
Solute carrier family 17 (sodium-dependent inorganic phosphate cotransporter) member 6	*Slc17a6*	−14.95	0.0037
Serine (or cysteine) peptidase inhibitor, clade B, member 1b	*Serpinb1b*	−12.79	0.0040
Peripherin	*Prph*	−12.69	0.00009
Neurofilament, medium polypeptide	*Nefm*	−11.77	0.0010
Neurofilament, heavy polypeptide	*Nefh*	-11.42	0.0079
Synuclein, gamma (breast cancer-specific protein 1)	*Sncg*	−10.76	0.0023
POU class 4 homeobox 2	*Pou4f2*	−7.30	0.0088
Neurofilament, light polypeptide	*Nefl*	−6.79	0.0027
Serine (or cysteine) proteinase inhibitor, clade B, member 1a	*Serpinb1a*	−4.94	0.0028
Tubulin polymerization-promoting protein family member 3	*Tppp3*	−4.56	0.0013
Transmembrane protein 163	*Tmem163*	−4.32	0.0010
Thy-1 cell surface antigen	*Thy1*	−3.88	0.0272
Major facilitator superfamily domain containing 6	*Mfsd6*	−3.73	0.0008
Chemokine (C-X-C motif) ligand 13	*Cxcl13*	−3.05	0.0142
Sodium channel, voltage-gated, type IV, beta	*Scn4b*	−2.91	0.0050
ISL LIM homeobox 2	*Isl2*	−2.90	0.0067
Sodium channel, voltage-gated, type I, beta	*Scn1b*	−2.83	0.033
Calpain 1, (mu/I) large subunit	*Capn1*	−2.79	0.0149
RAS guanyl releasing protein 2 (calcium and DAG-regulated)	*Rasgrp2*	−2.69	0.0153
Sodium channel, voltage-gated, type II, alpha 1	*Scn2a1*	−2.59	0.0301
Regulator of G-protein signaling 4	*Rgs4*	−2.48	0.0052
Visinin-like 1	*Vsnl1*	−2.45	0.0212
Synaptotagmin II	*Syt2*	−2.43	0.0250
NEL-like 2 (chicken)	*Nell2*	−2.40	0.0006
L1 cell adhesion molecule	*L1cam*	−2.35	0.0466
Plastin 3	*Pls3*	−2.29	0.0141
F-box protein 2	*Fbxo2*	−2.26	0.0041
Complexin 1	*Cplx1*	−2.25	0.0393
ELAV (embryonic lethal, abnormal vision, *Drosophila*)-like 2 (Hu antigen B)	*Elavl2*	−2.21	0.0221
Microsomal glutathione S-transferase 3	*Mgst3*	−2.19	0.0044
N-acetyltransferase 8-like	*Nat8l*	−2.17	0.0155
Annexin A6	*Anxa6*	−2.10	0.0047
Rho GTPase activating protein 32	*Arhgap32*	−2.08	0.0288
Leucine-rich repeat LGI family, member 3	*Lgi3*	−2.08	0.0132
Tyrosine 3-monooxygenase/tryptophan 5- monooxygenase activation protein, eta polypeptide	*Ywhah*	−2.05	0.0008
Ly6/neurotoxin 1	*Lynx1*	−2.03	0.0070
gb|Rattus norvegicus similar to glyceraldehyde-3-phosphate dehydrogenase (LOC304769), mRNA [XM_222600]	*XM_222600*	−2.00	0.0275
Receptor (G protein-coupled) activity modifying protein 3	*Ramp3*	−1.98	0.0331
FXYD domain-containing ion transport regulator 7	*Fxyd7*	−1.95	0.0023
Sulfotransferase family 4A, member 1	*Sult4a1*	−1.93	0.0115
Tubulin, beta 3 class III	*Tubb3*	−1.93	0.0163
Internexin neuronal intermediate filament protein, alpha	*Ina*	−1.93	0.0163
Parvalbumin	*Pvalb*	−1.91	0.0048
Potassium voltage-gated channel, Shal-related subfamily, member 2	*Kcnd2*	−1.89	0.0406
Heme binding protein 2	*Hebp2*	−1.89	0.0306
Pannexin 2	*Panx2*	−1.89	0.0050
Gremlin 2	*Grem2*	−1.86	0.0147
Prostaglandin F2 receptor negative regulator	*Ptgfrn*	−1.82	0.0021
Tropomodulin 4	*Tmod4*	−1.81	0.0412

**Table 2 genes-12-00403-t002:** Top 50 upregulated genes in ONC versus CTR.

Gene Name	Gene Symbol	Fold Change	*p*-Value
Cd74 molecule, major histocompatibility complex, class II invariant chain	*Cd74*	25.60	0.0030
tc|HG2A_RAT (P10247) H-2 class II histocompatibility antigen gamma chain (MHC class II-associated invariant chain) (Ia antigen-associated invariant chain) (Ii) (CD74 antigen), partial (31%) [TC588776]	*TC588776*	11.55	0.0100
Follistatin-like 3 (secreted glycoprotein)	*Fstl3*	5.31	0.0176
Serpin peptidase inhibitor, clade G, member 1	*Serping1*	3.49	0.0031
Eph receptor A2	*Epha2*	3.31	0.0061
Ceruloplasmin (ferroxidase)	*Cp*	3.29	0.0086
ADAM metallopeptidase with thrombospondin type 1 motif, 1	*Adamts1*	3.26	0.0097
Chitinase 3-like 1 (cartilage glycoprotein-39)	*Chi3l1*	2.89	0.0051
Cysteine and glycine-rich protein 3 (cardiac LIM protein)	*Csrp3*	2.69	0.0271
Leucine rich repeat containing 15	*Lrrc15*	2.61	0.0036
Protein tyrosine phosphatase, receptor type, O	*Ptpro*	2.58	0.0031
Keratin 19	*Krt19*	2.57	0.0013
Solute carrier family 17 (anion/sugar transporter), member 5	*Slc17a5*	2.54	0.0175
Scavenger receptor class A, member 3	*Scara3*	2.47	0.0086
Ellis van Creveld syndrome 2 homolog (human)	*Evc2*	2.47	0.0449
Cysteine and glycine-rich protein 2	*Csrp2*	2.44	0.0121
Complement component 1, s subcomponent	*C1s*	2.43	0.0157
S100 calcium binding protein A3	*S100a3*	2.42	0.0366
Complement component 4A (Rodgers blood group)	*C4a*	2.41	0.0118
PR domain containing 9	*Prdm9*	2.36	0.0225
Unknown	*A_64_P105338*	2.35	0.0068
Endothelin converting enzyme-like 1	*Ecel1*	2.34	0.0199
TIMP metallopeptidase inhibitor 1	*Timp1*	2.33	0.0115
Complement factor B	*Cfb*	2.26	0.0046
Transmembrane protein 176A	*Tmem176a*	2.19	0.0108
Solute carrier family 6 (neurotransmitter transporter, creatine), member 8	*Slc6a8*	2.18	0.0179
Janus kinase 3	*Jak3*	2.17	0.0144
Complement component 3	*C3*	2.17	0.0104
Interferon regulatory factor 2 binding protein-like	*Irf2bpl*	2.16	0.0412
SMAD family member 1	*Smad1*	2.12	0.0045
Aminoadipate aminotransferase	*Aadat*	2.09	0.0316
Plexin D1	*Plxnd1*	2.09	0.0250
Transmembrane BAX inhibitor motif containing 1	*Tmbim1*	2.08	0.0143
S100 calcium-binding protein A4	*S100a4*	2.03	0.0038
EGF-containing fibulin-like extracellular matrix protein 1	*Efemp1*	2.02	0.0229
Interferon induced transmembrane protein 3	*Ifitm3*	2.00	0.0251
Interferon, alpha-inducible protein 27-like 2B	*Ifi27l2b*	1.99	0.0214
Guanylate binding protein 2, interferon-inducible	*Gbp2*	1.96	0.0295
Ectonucleoside triphosphate diphosphohydrolase 2	*Entpd2*	1.96	0.0153
Interferon-induced protein 44-like	*Ifi44l*	1.95	0.0349
Lipopolysaccharide-induced TNF factor	*Litaf*	1.94	0.0245
Connective tissue growth factor	*Ctgf*	1.94	0.0261
Notch1	*Notch1*	1.94	0.0087
Cytochrome b-245, alpha polypeptide	*Cyba*	1.93	0.0306
SP110 nuclear body protein	*Sp110*	1.93	0.0346
biogenesis of lysosomal organelles complex-1, subunit 3	*Bloc1s3*	1.91	0.0152
Tachykinin receptor 1	*Tacr1*	1.91	0.0335
Coiled-coil domain containing 87	*Ccdc87*	1.90	0.0331
Ataxin 7-like 2	*Atxn7l2*	1.90	0.0320
G protein-coupled receptor 37	*Gpr37*	1.87	0.0286

**Table 3 genes-12-00403-t003:** Selected genes with altered expression in retinas ONC vs. CTR as confirmed by RT-qPCR.

	Microarray			qPCR	
**Gene Name**	**Pathway**	**ONC vs. CTR**	***p*-Value**	**ONC vs. CTR**	***p*-Value**
*Cd74*	N/A	25.60	0.003	28.56	0.010
*C3*	Complement Cascade	2.17	0.010	3.242	0.046
*Cyba*	Oxidative stress	1.94	0.03	1.41	0.01
*Fyn*	Kit receptor Signaling	1.63	0.045	1.47	0.04
*Tubb3*	RGC marker	−1.94	0.016	−3.93	0.004
*Nefm*	RGC marker	−11.78	0.001	−19,27	<0.0001
*Nell2*	RGC marker	−2.41	0.0006	−4.15	0.0012

**Table 4 genes-12-00403-t004:** RGC markers downregulated in ONC versus CTR.

Gene Name	Gene Symbol	Fold Change	*p*-Value
Neuritin 1	*Nrn1*	−18.82	0.0080
Solute carrier family 17 (sodium-dependent inorganic phosphate cotransporter) member 6	*Slc17a6*	−14.95	0.0037
Peripherin	*Prph*	−12.07	0.00009
Neurofilament, medium polypeptide	*Nefm*	−11.77	0.001
Neurofilament, heavy polypeptide	*Nefh*	−11.42	0.0079
Synuclein, gamma (breast cancer-specific protein 1)	*Sncg*	−10.76	0.0023
POU class 4 homeobox 2	*Pou4f2*	−7.30	0.0088
Neurofilament, light polypeptide	*Nefl*	−6.79	0.0027
Tubulin polymerization-promoting protein family member 3	*Tppp3*	−4.56	0.0013
Thy-1 cell surface antigen	*Thy1*	−3.88	0.0272
Regulator of G-protein signaling 4	*Rgs4*	−2.48	0.005
Visinin-like 1	*Vsnl1*	−2.46	0.021
Synaptogmin 2	*Syt2*	−2.43	0.025
F-box protein 2	*Fbxo2*	−2.27	0.004
Tyrosine-3-monooxygenase/tryptophan 5-monooxygenase activation protein, eta polypeptide	*Ywhah*	−2.05	0.0008
Tubulin, beta 3 class III	*Tubb3*	−1.93	0.016
Pannexin 2	*Panx2*	−1.89	0.005

**Table 5 genes-12-00403-t005:** Enrichment analysis with genes that had a dramatic change in their GES after ONC.

Gene Ontology	Number of Genes	Genes
Biological Process		
Lens development in camera-type eye	4	*Crygb*, *Crygc*, *Crygd*, and *Crygs*
Sensory organ development	7	*Cryba4*, *Crybb2*, *Crygb*, *Crygc*, *Crygd*, *Crygs*, and *Krt13*
Camera-type eye development	6	*Cryba4*, *Crybb2*, *Crygb*, *Crygc*, *Crygd*, and *Crygs*
Molecular Function		
Structural constituent of eye lens	9	*Cryaa*, *Cryba4*, *Crybb2*, *Crybb3*, *Crygb*, *Crygc*, *Crygd*, *Crygs*, and *Lim2*
Structural molecule activity	13	*Anxa1*, *Cryaa*, *Cryba4*, *Crybb2*, *Crybb3*, *Crygb*, *Crygc*, *Crygd*, *Crygs*, *Krt12*, *Krt13*, *Krt80*, and *Lim2*

**Table 6 genes-12-00403-t006:** Pathvisio analysis of upregulated pathways in ONC versus CTR retinas.

Pathway	Positive (r)	Measured (n)	Total	%	Z Score	*p*-Value (Permuted)
Complement activation, classical pathway	3	15	18	20.00%	6.27	0
Complement and coagulation cascades	5	56	63	8.93%	4.96	0
Delta-Notch signaling pathway	4	72	82	5.56%	3.13	0.009
Oxidative stress	2	26	28	7.69%	2.81	0.031
Notch signaling pathway	2	32	45	6.25%	2.41	0.017
Kit receptor signaling pathway	3	64	68	4.69%	2.34	0.04

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
