# Peer review of "Retinal Genomic Fabric Remodeling after Optic Nerve Injury"

_genes, 2021, doi:10.3390/genes12030403_

Round 1
Reviewer 1 Report
Title: Retinal genomic fabrics remodeling after optic nerve injury
The manuscript by Victorino et al. investigates the expression regulation of individual genes involved in RGC degeneration and remodeling of the retinal tissue, using an animal model of glaucoma (optic nerve crush). The manuscript is well written and the results of the study are potential of interest. Even though, to be suitable to be published, there are still some minor revisions to address. These issues are outlined below.
Please change “complement cascade” with “Complement Cascade” (capital letters) in line 536.
Please insert the letter A and B to indicate Control and Crush in figure 6, since you described in the text the figures as Figure 6B and 6A (lines 359 -360). The same for figure 7 and also, you need to add the letter A in line 374. The same for figure 8 and also, you need to add the letter A in line 383.
Please correct Table S7(line 475) to Table 7, since there is no Table S7.
In Table 1,2,4, please align “symbol” towards “Gene” because at now it seems that the terms indicate separated columns in the table.
Please choose one form between “p-value” and “p value to indicate it (please check in the text and in the tables).
Please change the reference in line 447 from the Author’s name to the number.
Author Response
The manuscript by Victorino et al. investigates the expression regulation of individual genes involved in RGC degeneration and remodeling of the retinal tissue, using an animal model of glaucoma (optic nerve crush). The manuscript is well written and the results of the study are potential of interest. Even though, to be suitable to be published, there are still some minor revisions to address. These issues are outlined below.
Please change “complement cascade” with “Complement Cascade” (capital letters) in line 536.
Please insert the letter A and B to indicate Control and Crush in figure 6, since you described in the text the figures as Figure 6B and 6A (lines 359 -360). The same for figure 7 and also, you need to add the letter A in line 374. The same for figure 8 and also, you need to add the letter A in line 383.
Please correct Table S7(line 475) to Table 7, since there is no Table S7.
In Table 1,2,4, please align “symbol” towards “Gene” because at now it seems that the terms indicate separated columns in the table.
Please choose one form between “p-value” and “p value to indicate it (please check in the text and in the tables).
Please change the reference in line 447 from the Author’s name to the number.
Response: We thank this reviewer for the positive feedback and the suggestions to improve our manuscript. All minor issues herein indicated were fixed as requested.
Reviewer 2 Report
Why is part of the text highlighted in yellow?
Title
I believe that the term ‘genomic fabrics’ is not common knowledge among neuroscientists and therefore recommend rephrasing the title “Retinal genomic fabrics remodeling after optic nerve injury”.
Abstract
- "Thus, for the 17,657 quantified genes our study generated a total of 155,911,310 values to analyze that is 8,830x more data per condition than a traditional transcriptomic analysis, ONC led to a 57% reduction in RGC numbers as detected by retrograde labeling with DiI.” This should be 3 sentences. The information on RGC loss seems irrelevant here.
- “This deep bioinformatic study provided novel insights beyond the regulation of individual gene expression and disclosed changes in the control of expression by Complement cascade and Notch signaling functional pathways important for both RGC degeneration and remodeling of the retinal tissue after ONC.” I’m not sure whether the control of expression is “changed by” Complement cascade and Notch signaling. Please rephrase this sentence to make more clear. Use present tense.
Introduction
- “ONC is an acute, mechanical injury to the nerve, that leads to gradual RGC apoptosis and have been widely utilized to examine glaucomatous disease pathophysiology.” Have should be has.
- Complement Cascade, Notch Signaling Pathway, Oxidative Stress and Kit receptor Signaling pathways should not be written with capital letters.
Methods
- I’m confused about the n numbers stated in the methods sections. First, it’s written “The rats were divided into two experimental groups: optic nerve crush (n=24, hereafter denoted by ONC) and control, sham operated, (n=19, denoted by CTR).” Next, the authors say “For each experimental group (CTR or ONC), four separated retinas were profiled individually.” and “To validate selected genes, a new set of animals (n = 4 per group) was used for qRT-PCR.” This adds up to a total of 8 rats per group. Where did all the other samples go?
- “For each experimental group (CTR or ONC), four separated retinas were profiled individually.” is written twice in the same section.
- It is unconventional to label the RGCs with DiI so many weeks before the ONC and sacrifice. By retrograde labeling the RGCs with DiI 7 weeks before ONC, labelled cells will be phagocytosed by microglia and macrophages upon ONC, and hence these non-RGC cell type will become DiI as well. One should do the retrograde labelling after the ONC, just prior to sacrifice. The illustrations in figure 1 indeed reveal DiI-labelled structures that lack a typical spheroid shape – as is usually seen for retrogradely labeled RGCs – and rather show irregular, diffuse DiI signal that may represent cellular debris or DiI engulfed by immune cells. Ion figure 1C, t is absolutely unclear to me how this signal can be interpreted and counted as cells. As such, the quantification of RGC loss seems unreliable. The authors should apply DiI labeling later, or use immunostaining for an RGC-specific marker such as Brn3a or RBPMS.
Results
- “Quantitative RT-PCR validated two of the highest up-regulated genes (Cd74 and C3) and three markers of RGCs, along with their respective fold-change alteration in expression, compared with CTR retinas, were consistent with the results of the arrays (Table 3).” Which three RGC markers were confirmed by qPCR? What was the fold changes in the microarray versus the qPCR? Rephrase into two sentences. The note about the RGC markers should be transferred to the next alinea.
- “Consistent with previous studies [15-18; 28], the group of 50 most up-regulated genes (Table 2) include inflammatory response-related genes, ...” Include should includes.
- The validation studies of the microarray data are very limited. First, expression data is only validated at the mRNA level. Second, only one gene for the complement cascade, oxidative stress and Kit pathway have been validated via qPCR. I recommend validating also additional key genes as well as genes related to Notch signalling.
Discussion
- “Although protein-protein interaction determined with STRING may be well described prediction of real interactions through data mining in genome wide datasets (Szklarczyk et al., 2019),...” Rephrase.
- A whole paragraph is dedicated to crystallin, while crystallin are not mentioned in the results section.
- Whether or not the retina can be considered immune-privileged, is hotly debated. I recommend removing this term from the discussion, as it is not strictly necessary and controversial.
- I disagree with the statement “To date, inflammation has been considered secondary to the tissue damage, and is now thought to be part of a protective response of the immune system.” First, there is no conclusive evidence to suggest that inflammation is a consequence rather than a cause in glaucomatous neurodegeneration - I believe it is a very complex issue, with a truth probably lying somewhere in between – nor is there any evidence to say that it is protective. In fact, many papers suggest the opposite – again, this probably depends on the very complex interplay of timing, type of injury,...
- “At 14 days after ONC, the up-regulation of both Notch and deltex2 suggests that Notch signaling may act upon retinal Müller cells via a non-canonical pathway, thereby positively influencing RGC degeneration.” How does Notch signaling confer a positive effect on RGC degeneration?
- The discussion now reads as a series of paragraphs that each dig deeper into one of the pathways identified in this study. It would be informative to bring all these findings together in one parapraph that integrates these data into a biological story, e.g. how are inflammation, oxidative, stress, complement,... linked, how does Notch and Kit signalling related to these, and how do they link to RGC death and survival.
Conclusions
- “In summary, our data indicate that ONC induces a robust, long-lasting alteration of two main pathways, including a significant increase in the expression and coordination of inflammation-related genes.” Please state which 2 pathways.
Figures & tables
The manuscript contains a very large number of tables and figures. I recommend moving some of them to the supplement.
Author Response
Reviewer #2
Why is part of the text highlighted in yellow?
Response: Parts of the text are in yellow since this was the R3 submission of this manuscript. We were asked to perform additional analyses, including qRT-PCRs to validate differentially expressed genes in ONC compared to CTR. Such new data and information were also highlighted in the re-submitted paper (R3) which are now modified following the suggestions made by reviewers #1 and #2.
Title
I believe that the term ‘genomic fabrics’ is not common knowledge among neuroscientists and therefore recommend rephrasing the title “Retinal genomic fabrics remodeling after optic nerve injury”.
Response: Since this special issue of this journal is named “Genomic Fabric Remodeling in Neurological Disorders” we feel that this title falls within the scope of the issue.
Abstract
"Thus, for the 17,657 quantified genes our study generated a total of 155,911,310 values to analyze that is 8,830x more data per condition than a traditional transcriptomic analysis, ONC led to a 57% reduction in RGC numbers as detected by retrograde labeling with DiI.” This should be 3 sentences. The information on RGC loss seems irrelevant here.
Response: We corrected this as suggested by the reviewer.
“This deep bioinformatic study provided novel insights beyond the regulation of individual gene expression and disclosed changes in the control of expression by Complement cascade and Notch signaling functional pathways important for both RGC degeneration and remodeling of the retinal tissue after ONC.” I’m not sure whether the control of expression is “changed by” Complement cascade and Notch signaling. Please rephrase this sentence to make more clear. Use present tense.
Response: We have changed “provided” and “disclosed” to present tense and have changed this sentence to read: “This deep bioinformatic study provides novel insights beyond the regulation of individual gene expression and discloses changes in the control of expression of Complement Cascade and Notch Signaling functional pathways that may be relevant for both RGC degeneration and remodeling of the retinal tissue after ONC.”
Introduction
“ONC is an acute, mechanical injury to the nerve, that leads to gradual RGC apoptosis and have been widely utilized to examine glaucomatous disease pathophysiology.” Have should be has.
Response: This was changed as requested.
Complement Cascade, Notch Signaling Pathway, Oxidative Stress and Kit receptor Signaling pathways should not be written with capital letters.
Response: We have used capital letters for molecular pathways as requested by previous reviewers and have tried to standardize this for all of them throughout the manuscript.
Methods
I’m confused about the n numbers stated in the methods sections. First, it’s written “The rats were divided into two experimental groups: optic nerve crush (n=24, hereafter denoted by ONC) and control, sham operated, (n=19, denoted by CTR).” Next, the authors say “For each experimental group (CTR or ONC), four separated retinas were profiled individually.” and “To validate selected genes, a new set of animals (n = 4 per group) was used for qRT-PCR.” This adds up to a total of 8 rats per group. Where did all the other samples go?
Response: You are correct, the remaining retinas (11 CTR and 16 ONC) were used for RGC quantification with DiI. This information was added to the Methods section.
“For each experimental group (CTR or ONC), four separated retinas were profiled individually.” is written twice in the same section.
Response: Thank you for pointing this out. The repeated sentence was removed in the revised manuscript.
It is unconventional to label the RGCs with DiI so many weeks before the ONC and sacrifice. By retrograde labeling the RGCs with DiI 7 weeks before ONC, labelled cells will be phagocytosed by microglia and macrophages upon ONC, and hence these non-RGC cell type will become DiI as well. One should do the retrograde labelling after the ONC, just prior to sacrifice. The illustrations in figure 1 indeed reveal DiI-labelled structures that lack a typical spheroid shape – as is usually seen for retrogradely labeled RGCs – and rather show irregular, diffuse DiI signal that may represent cellular debris or DiI engulfed by immune cells. Ion figure 1C, it is absolutely unclear to me how this signal can be interpreted and counted as cells. As such, the quantification of RGC loss seems unreliable. The authors should apply DiI labeling later, or use immunostaining for an RGC-specific marker such as Brn3a or RBPMS.
Response: Until one week after birth, the superior colliculus in rats is not covered by the cortex and is easily accessed for DiI injection. Unfortunately, we do not have a stereotaxic to inject adult rats. Moreover, it is well known that RGC remain labeled with DiI for periods of up to 9 months without apparent leakage of the tracer to other retinal cells. In addition, DiI labeling persisted in the somata of more than 80% of axotomized RGCs whose contact with the source of label had been interrupted for 3 months (DOI: 10.1016/0014-4886(88)90081-7). These results support our protocol of analysis seven weeks after DiI injection.
Related with BRN3a, since its characterization as a marker of RGCs, it was shown that in rats, the expression of Brn3a in the injured retinas decreased with death of RGCs, and there is a down-regulation of this protein in the surviving ones (DOI: 10.1371/journal.pone.0049830). Moreover, Brn3a down-regulation signals apoptotic RGC death (DOI: 10.1167/iovs.15-17841) and correlates with the course of RGC loss, that is, the quicker the model of RGC death, the higher the down-regulation of Brn3a per RGC. In fact, we verified down-regulation of Brn3a expression in our arrays in ONC when compared to CTR retinas (POU class 4 homeobox 2, -7.3x, p=0.0088), which corroborates RGC loss revealed by DiI injection.
Intraorbital optic nerve crush (ONC) is a clean and reproducible model with relatively little inter-animal variability as assessed by counting retinal ganglion cells (Vidal-Sanz et al., 1987; Peinado-Ramón et al., 1996; Sobrado-Calvo et al., 2007; Parrilla-Reverter et al., 2009b; Sánchez-Migallón et al., 2011, 2016; Nadal-Nicolás et al., 2015; Rovere et al., 2016). Our result of ONC, assessed by counting retrograde labeled retinal ganglion cells, did not present high variability and it was very consistent with the literature.
RBPMS is a great alternative to count RGCs however we did not have a RBPMS antibody.
Results
“Quantitative RT-PCR validated two of the highest up-regulated genes (Cd74 and C3) and three markers of RGCs, along with their respective fold-change alteration in expression, compared with CTR retinas, were consistent with the results of the arrays (Table 3).” Which three RGC markers were confirmed by qPCR? What was the fold changes in the microarray versus the qPCR? Rephrase into two sentences. The note about the RGC markers should be transferred to the next alinea.
Response: We have rephrased the description of qPCR data, which are now described in a separate paragraph as follows: “Quantitative RT-PCR was used to validate two of the highest up-regulated genes (Cd74 and C3) and three markers of RGCs (Tubb3, Nefm, Nell2) that were found to be down-regulated in ONC retinas as compared to CTR (Table 3). Cd74 and C3 were found to be up-regulated by qPCR by 28.56- and 3.24-fold, respectively. Tubb3, Nefm and Nell2 were down-regulated by -3.93-, -19.27- and -4.15-fold in ONC retinas when compared to CTR ones, consistent with the results of the arrays (Table 3).”.
“Consistent with previous studies [15-18; 28], the group of 50 most up-regulated genes (Table 2) include inflammatory response-related genes, ...” Include should includes.
Response: This was corrected in the revised manuscript, thank you.
The validation studies of the microarray data are very limited. First, expression data is only validated at the mRNA level. Second, only one gene for the complement cascade, oxidative stress and Kit pathway have been validated via qPCR. I recommend validating also additional key genes as well as genes related to Notch signaling.
Response: We recognize that it is important to validate gene expression differences using independent methods, using such methods as RT-qPCR and immunodetection, and in numerous previous studies we have demonstrated correspondence using multiple methods.
However, identical results are not expected because each method is subject to additional sources of technical noise (see Athanasiou et al., 2020) and the distinct samples required for these analyses introduces additional sources of biological variability. For these reasons we have chosen to validate expression differences that are not only highly significant but also represent high abundance transcripts, which are likely to overcome technical noise and biological variability. This was performed after the initial submissions (R0 and R1) and have included all the qPCR validations (Cd74, C3, Cyba, Fyn, Tubb3, Nefm, Nell2) as requested after R2 submission.
It is important to highlight that the original scope of this work is the deep bioinformatic study that was employed, which is in consonance with the theme of this special issue of Genes (Genomic Fabric Remodeling in Neurological Disorders).
- Athanasiou AT, Nussbaumer T, Kummer S, Hofer M, Johnston IG, Staltner M, Allmer DM, Scott MC, Vogl C, Fenger JM, Modiano JF, Walter I, Steinborn R. S100A4 mRNA-protein relationship uncovered by measurement noise reduction. J Mol Med (Berl). 2020 Apr 15. doi: 10.1007/s00109-020-01898-8.
Discussion
“Although protein-protein interaction determined with STRING may be well described prediction of real interactions through data mining in genome wide datasets (Szklarczyk et al., 2019),...” Rephrase.
Response: This sentence was rephrased to read: “Although protein-protein interaction determined with STRING (based on data mining in genome wide datasets) may be a good predictor of real interactions (Szklarczyk et al., 2019), we deepened our analyses with additional bioin-formatics and mathematical modeling of gene networks of ONC retinas….”.
A whole paragraph is dedicated to crystallin, while crystallin are not mentioned in the results section.
Response: We agree that the description of changes in crystallin GES was missing from the Results section. Therefore, the initial sentence was moved to the Results section and at the Discussion we changed to read: “The crystallin genes were found to have relevant changes in their GES, thus acquiring a more unstable expression profile. Crystallins have chaperone-like properties involved in an increase of cellular resistance to stress-induced apoptosis….”.
Whether or not the retina can be considered immune-privileged, is hotly debated. I recommend removing this term from the discussion, as it is not strictly necessary and controversial.
Response: The term “immune-privileged” was removed and this sentence was changed to read: “Such activation, specifically including C1q, C3, and Cfi [20], suggests that the innate immune system plays an important role in retinal immune response through glia cells, and specifically….”,
I disagree with the statement “To date, inflammation has been considered secondary to the tissue damage, and is now thought to be part of a protective response of the immune system.” First, there is no conclusive evidence to suggest that inflammation is a consequence rather than a cause in glaucomatous neurodegeneration - I believe it is a very complex issue, with a truth probably lying somewhere in between – nor is there any evidence to say that it is protective. In fact, many papers suggest the opposite – again, this probably depends on the very complex interplay of timing, type of injury,...
Response: We agree that this topic is controversial and was not the main focus of this manuscript. Therefore, we removed this sentence to improve readability.
“At 14 days after ONC, the up-regulation of both Notch and deltex2 suggests that Notch signaling may act upon retinal Müller cells via a non-canonical pathway, thereby positively influencing RGC degeneration.” How does Notch signaling confer a positive effect on RGC degeneration?
Response: Many thanks for the question. Notch signaling mediates many different intercellular communication events, influencing retinal Muller cells, ganglion cells and microglial cells. It is known that Notch-1 signaling in neurons during ischemic conditions may enhance apoptotic signaling cascades, while activation of notch-1 in microglial cells and leukocytes may exacerbate inflammatory processes that contribute to neuronal death (Eagar et al., 2004). Also, several studies found that notch signaling serves important functions in the regulation of neurite outgrowth and maintenance. Both Sestan et al., (1999) and Berezovska et al., (1999) found that notch signaling influence dendritic morphology: while activation inhibits neurite outgrowth or causes their retraction, inhibition promotes neurite extension. All these information’s, together with our results suggest that Notch signaling may confer a positive effect on RGC degeneration through an exacerbation of inflammatory processes and also promoting retinal tissue remodeling, by altering dendritic morphology. We have added this paragraph to the Discussion section.
- Eagar TN, Tang Q, Wolfe M, He Y, Pear WS, Bluestone JA. Notch 1 signaling regulates peripheral T cell activation. Immunity. 2004, 20 (4): 407-15. doi:10.1016/s1074-7613(04)00081-0.
- Sestan N, Artavanis-Tsakonas S, Rakic P. Contact-dependent inhibition of cortical neurite growth mediated by notch signaling. Science. 1999, 286 (5440): 741-6. doi: 10.1126/science.286.5440.741
- Berezovska O, McLean P, Knowles R, Frosh M, Lu FM, Lux SE, Hyman BT. Notch1 inhibits neurite outgrowth in postmitotic primary neurons. Neuroscience. 1999, 93 (2): 433-9. doi: 10.1016/s0306-4522(99)00157-8.
The discussion now reads as a series of paragraphs that each dig deeper into one of the pathways identified in this study. It would be informative to bring all these findings together in one parapraph that integrates these data into a biological story, e.g. how are inflammation, oxidative, stress, complement,... linked, how does Notch and Kit signalling related to these, and how do they link to RGC death and survival.
Response: Thanks for this comment. We added the paragraph below in order to integrate the data into a biological story, as suggested.
Oxidative stress and inflammation are closely related pathophysiological processes, one of which can be easily induced by another. Thus, both processes are simultaneously found in many chronical pathological conditions, such as glaucoma (REF1). The immune activity is a necessary intrinsic response to promote tissue cleaning, healing, and regeneration process after injury (REF2). However, failure of its regulation due to increasing risk factors, turn the beneficial potential into potentiation of the injury process. The mechanical injury together with hypoxia, both promoted by the ONC could be the triggers for the activation of inflammation, oxidative stress and metabolism alterations. Notch-1, c-Kit and complement signaling are known as linkers between inflammation, metabolism, oxidative stress and dendritic plasticity, acting into RGC and Müller cells (REF3-8). Our results support the hypothesis that not only the events intrinsic to RGCs, but also environmental signals from other cells, such as Müller cells, are critical for cell death stimuli, and RGC-glia interactions are critically important for different aspects of RGC neurodegeneration. Improvement of the current understanding of the role of RGC-glia interaction, mitochondria, and the immune system in neurodegeneration will potentiate the development of innovative treatment methods for neuroprotection.
REF1: Vernazza S, Tirendi S, Bassi AM, Traverso CE, Saccà SC. Neuroinflammation in Primary Open-Angle Glaucoma. J Clin Med. 2020, 9(10): 3172. doi: 10.3390/jcm9103172.
REF2: Yong HYF, Rawji KS, Ghorbani S, Xue M, Yong VW. The benefits of neuroinflammation for the repair of the injured central nervous system. Cell Mol Immunol. 2019, 16(6): 540-546. doi: 10.1038/s41423-019-0223-3.
REF3: Clark SJ, Bishop PN. The eye as a complement dysregulation hotspot. Semin Immunopathol. 2018, 40(1): 65-74. doi: 10.1007/s00281-017-0649-6.
REF4: Tezel G, Yang X, Luo C, et al. Oxidative stress and the regulation of complement activation in human glaucoma. Invest Ophthalmol Vis Sci. 2010, 51(10): 5071-5082. doi:10.1167/iovs.10-5289.
REF5: Tezel G; Fourth ARVO/Pfizer Ophthalmics Research Institute Conference Working Group. The role of glia, mitochondria, and the immune system in glaucoma. IOVS 2009, 50(3):1001-12. doi: 10.1167/iovs.08-2717.
REF6: Lathia JD, Mattson MP, Cheng A. Notch: from neural development to neurological disorders. J Neurochem. 2008, 107(6):1471-1481. doi:10.1111/j.1471-4159.2008.05715.x
REF7: Hess C, Kemper C. Complement-Mediated Regulation of Metabolism and Basic Cellular Processes. Immunity. 2016, 45(2):240-54. doi: 10.1016/j.immuni.2016.08.003.
REF8: Ries A, Goldberg JL, Grimpe B. A novel biological function for CD44 in axon growth of retinal ganglion cells identified by a bioinformatics approach. J Neurochem. 2007;103(4):1491-1505. doi:10.1111/j.1471-4159.2007.04858.x
Conclusions
“In summary, our data indicate that ONC induces a robust, long-lasting alteration of two main pathways, including a significant increase in the expression and coordination of inflammation-related genes.” Please state which 2 pathways.
Response: We have changed this to read: “In summary, our data indicate that ONC induces a robust, long-lasting alteration of at least two main pathways (Complement Cascade and Notch Signaling Pathway), including…”
Figures & tables
The manuscript contains a very large number of tables and figures. I recommend moving some of them to the supplement.
Response: We agree with the reviewer and have transferred Figures 6 (now Figure S2), 7 (Figure S3) and 8 (Figure S4) and Tables 5 (now Table S1) and 6 (Table S2) to the supplement. The references to these figures and tables have been modified throughout the manuscript.
Round 2
Reviewer 2 Report
I first want to apologise for the comments related to 'Genomic Fabric Remodeling' and the validation via qPCR. I was unaware that this is a resubmission and therefore did not know what changes had already been requested/instructions were given for the special issue.
The authors have addressed most of my concerns, and the quality of the manuscript did improve. However, I have two remaks left:
- I agree with the authors that DiI labeling stays in the RGCs for a long time after retrograde labeling. This is not my concern. My concern is that you have been counting non-RGCs because phagocytic cells are engulfing death RGCs and thereby become DiI-positive too. This phenomenon is well known in the glaucoma research field. It should be tackled by (i) alternative labeling strategies (later DiI injection or immunostainings for RGC markers) or (ii) a counting protocol that excludes these cells. Not having a stereotact or an antibody is not an excuse for conducting a low-quality experiment.
- In the new paragraph in de Discussion, you write "However, failure of its regulation due to increasing risk factors, turn the beneficial potential into potentiation of the injury process." I don't understand what you mean by "due to increasing risk factors". Isn't the inflammatory respons dysregulated to some extend. Of not, then why is there still RGC death and no recovery?
Author Response
The authors have addressed most of my concerns, and the quality of the manuscript did improve. However, I have two remarks left:
I agree with the authors that DiI labeling stays in the RGCs for a long time after retrograde labeling. This is not my concern. My concern is that you have been counting non-RGCs because phagocytic cells are engulfing death RGCs and thereby become DiI-positive too. This phenomenon is well known in the glaucoma research field. It should be tackled by (i) alternative labeling strategies (later DiI injection or immunostainings for RGC markers) or (ii) a counting protocol that excludes these cells. Not having a stereotact or an antibody is not an excuse for conducting a low-quality experiment.
Response: Thank you for this clarification. We now understood your concern, and it is about the quantification method of DiI staining of RGCs. You are absolutely right about the necessity of efforts to exclude microglial cells with translocated DiI. This is why we manually quantified DiI+ RGCs, based on their morphology, as we do it as a standard in our lab. In order to make it clear in the text, we added this information into the Materials and Methods section and we also added arrowheads into the Figure 1B to clarify the morphology of the DiI cells we considered as RGCs, for the quantification.
Our confocal images, both CTR and ONC, clearly show many DiI positive RGCs. As mentioned before DiI is a reliable tracer, that does not diffuse between cells, as occurs with Fluorogold, and is highly successful to retrolabel RGCs, showing the classical pattern of spread punctuated cytoplasmic staining. It is well known, as mentioned, that the engulfment of dead RGC makes it impossible to use automated quantification. Therefore, manual quantification is broadly accepted since microglial cells have very distinguished morphology and smaller nucleus compared with RGCs.
In the new paragraph in the Discussion, you write "However, failure of its regulation due to increasing risk factors, turn the beneficial potential into potentiation of the injury process." I don't understand what you mean by "due to increasing risk factors". Isn't the inflammatory response dysregulated to some extent. Of not, then why is there still RGC death and no recovery?
Response: We have changed this sentence to read “However, unbalanced or sustained immune response, turns the beneficial potential into potentiation of the injury process.”
This manuscript is a resubmission of an earlier submission. The following is a list of the peer review reports and author responses from that submission.
Round 1
Reviewer 1 Report
Reviewer’s Comments (genes-775202)
Title: Optic nerve crush activates the genomic fabrics of the Complement cascade and Delta-Notch signaling
The manuscript by Victorino et al. investigates the expression regulation of individual genes involved in RGC degeneration and remodeling of the retinal tissue, using an animal model of glaucoma (optic nerve crush). The manuscript is well written and the results of the study are potential of interest. Even though, to be suitable to be published, there are major and minor revisions to address. These issues are outlined below.
Major revision:
The authors should notice that Figure 7 and Figure 8 have been showed in the “Results” section without describing them nor in the “Results” neither in the “discussion” and “conclusion”. Only in lines 340 and 341, it is reported a brief description of the data shown in Figures 7 and 8: “Oxidative Stress and Kit Receptor Signaling Pathway (Figures 7 and 8).” Please, carefully check this part as well as the numbers of the figures that the authors report in the manuscript sections because many of them seem to be not in accordance with the sentences toward they are referred to (i.e. Lines 342-365: maybe should be Figure 4 and not Figure 6? Lines 380-390: maybe should be Figure 5 and 6 and not Figure 7?).
Minor revisions:
- After abbreviating for the first time a term, you should use the relative abbreviations.
Please check and correct “retinal ganglion cell” to “RGC” or “retinal ganglion cells“ to “RGCs” in lines: 231, 236, 425, 470, 492.
The same revision should be done for other terms such as: “optic nerve crush” to ONC (lines: 182, 322.); “relative expression variability” to “REV” (Line 316); “Gene Expression Stability score” to “GES”.
- Some titles of the figure legends are not in bold (lines: 279,404, 411). Further, only in some figure legends, the letters indicating a part of the figure are in bold as well as the brackets that surround them. Please choose one form (if needed check the journal author’s guidelines), then correct.
Other minor revisions:
- Line 41: RGC should be RGCs since is “cells” and not “cell”.
- Line 42: Please change “in vivo” with “in vivo” (italic font).
- Line 69: In this line, “subacute” phase is reported differently compared to line 65 (sub-acute). Please choose one form and correct.
- Line 72: In this line, “up-regulated” is reported differently compared to line 70 (upregulation). Please choose one form and correct.
- Line 76: Iis there an extended name for YY1? If yes, please add it.
- Line 100: Please change “ad libitum” with “ad libitum” (italic font)
- Line 134: What does OPC means? Please, provide the extended name.
- Line 173: Please correct “vs” with “vs” (italic font)
- Line 178: “cutoff” is reported differently compared to line 159, 161, 163 (cut-off). Please choose one form and correct.
- Line 184: In this line, “p value” is reported differently compared to line 159 (p-value). Please choose one form and correct.
- Line 189: GEO (Gene Expression Omnibus). Should be: Gene Expression Omnibus (GEO), as you wrote for all the other extended names/abbreviations. Please correct.
- Line 217: “Cd74” the number 4 is not in bold. Please correct.
- Line 238: Missing point “.” after “green”. Please correct.
- Figure 1 (c): What does “***” refer to? Please explain.
- Line 256, 273, 275, 276: Please delete bracket before the letter (d), (a), (b) and (c).
- Lines 273: Please in “Figure2(a))” add the space after “2”
- Line 297: Please delete double space after “score of 0.4”
- Lines 299; 301 and 303: Please correct “Figure 43(a))” with “Figure 3 a”. Delete brackets (b) and (c).
- Line 326: What is Panther? Should be better to cite it in the materials and methods.
- Line 336: “Pathvisio3” is reported differently from “PathVisio3” (also in line 429). Please, choose one form and correct.
- Line 367: pathway is not reported with p capital letter. Please correct.
- Line 372, 398; 408; 414 and 421: Please correct “versus” with “versus” (italic font)
- Line 373, 399, 409, 415, 422: What does “p” stand for? P-value or Pearson coefficient?
- Line 393: pathway is not reported with “p” capital letter. Please correct.
- Line 379: where is Table 6?
- Line 437: Muller is written wrong. Please correct.
- Line 458: Check if it is correct to indicate “(Figure 6A)” here.
- Line 478: Maybe the correct figure is the “Figure1”?
- Line 508: Please correct double brackets into single ones.
- Line 524: Maybe should be “Figure 4” and not Figure 7?
- Line 540: “notch” is reported without capital letter. Please correct.
- Line 551: ONC affects, the “s” is missing.
Author Response
The manuscript by Victorino et al. investigates the expression regulation of individual genes involved in RGC degeneration and remodeling of the retinal tissue, using an animal model of glaucoma (optic nerve crush). The manuscript is well written and the results of the study are potential of interest. Even though, to be suitable to be published, there are major and minor revisions to address. These issues are outlined below.
Response: We thank the reviewer for the positive feedback to our manuscript and have addressed most of their comments and issues that were raised in their report. All formatting issues were corrected as indicated by this reviewer and most of them are highlighted in yellow throughout the text. We did not highlight all the italicized gene names because they are so numerous and have obviously been revised.
Major revision:
The authors should notice that Figure 7 and Figure 8 have been showed in the “Results” section without describing them nor in the “Results” neither in the “discussion” and “conclusion”. Only in lines 340 and 341, it is reported a brief description of the data shown in Figures 7 and 8: “Oxidative Stress and Kit Receptor Signaling Pathway (Figures 7 and 8).” Please, carefully check this part as well as the numbers of the figures that the authors report in the manuscript sections because many of them seem to be not in accordance with the sentences toward they are referred to (i.e. Lines 342-365: maybe should be Figure 4 and not Figure 6? Lines 380-390: maybe should be Figure 5 and 6 and not Figure 7?).
Response: These Figures are now cited in the Results section. Citations of all figures were also revised for consistency throughout the manuscript.
Minor revisions: (all have now been corrected)
After abbreviating for the first time a term, you should use the relative abbreviations.
Please check and correct “retinal ganglion cell” to “RGC” or “retinal ganglion cells“ to “RGCs” in lines: 231, 236, 425, 470, 492.
The same revision should be done for other terms such as: “optic nerve crush” to ONC (lines: 182, 322.); “relative expression variability” to “REV” (Line 316); “Gene Expression Stability score” to “GES”.
Some titles of the figure legends are not in bold (lines: 279,404, 411). Further, only in some figure legends, the letters indicating a part of the figure are in bold as well as the brackets that surround them. Please choose one form (if needed check the journal author’s guidelines), then correct.
Other minor revisions:
Line 41: RGC should be RGCs since is “cells” and not “cell”.
Line 42: Please change “in vivo” with “in vivo” (italic font).
Line 69: In this line, “subacute” phase is reported differently compared to line 65 (sub-acute). Please choose one form and correct.
Line 72: In this line, “up-regulated” is reported differently compared to line 70 (upregulation). Please choose one form and correct.
Line 76: Iis there an extended name for YY1? If yes, please add it.
Line 100: Please change “ad libitum” with “ad libitum” (italic font)
Line 134: What does OPC means? Please, provide the extended name.
Line 173: Please correct “vs” with “vs” (italic font)
Line 178: “cutoff” is reported differently compared to line 159, 161, 163 (cut-off). Please choose one form and correct.
Line 184: In this line, “p value” is reported differently compared to line 159 (p-value). Please choose one form and correct.
Line 189: GEO (Gene Expression Omnibus). Should be: Gene Expression Omnibus (GEO), as you wrote for all the other extended names/abbreviations. Please correct.
Line 217: “Cd74” the number 4 is not in bold. Please correct.
Line 238: Missing point “.” after “green”. Please correct.
Figure 1 (c): What does “***” refer to? Please explain.
Response: This symbol “***” symbol referred to the statistical significance of this quantification and is now included in the figure legend.
Line 256, 273, 275, 276: Please delete bracket before the letter (d), (a), (b) and (c).
Lines 273: Please in “Figure2(a))” add the space after “2”
Line 297: Please delete double space after “score of 0.4”
Lines 299; 301 and 303: Please correct “Figure 43(a))” with “Figure 3 a”. Delete brackets (b) and (c).
Line 326: What is Panther? Should be better to cite it in the materials and methods.
Response: Panther is the acronym for Protein ANalysis THrough Evolutionary Relationships, a classification system that was used to analyze the enrichment of genes within a specific subset of genes. This description was added to the methodology section.
Line 336: “Pathvisio3” is reported differently from “PathVisio3” (also in line 429). Please, choose one form and correct.
Response: This was corrected to PathVisio3 throughout the manuscript.
Line 367: pathway is not reported with p capital letter. Please correct.
Line 372, 398; 408; 414 and 421: Please correct “versus” with “versus” (italic font)
Line 373, 399, 409, 415, 422: What does “p” stand for? P-value or Pearson coefficient?
Response: This “p” stands for p-value and was corrected in the text.
Line 393: pathway is not reported with “p” capital letter. Please correct.
Line 379: where is Table 6?
Response: Numbering of figures and tables was revised throughout the manuscript. We apologize for this mistake.
Line 437: Muller is written wrong. Please correct.
Line 458: Check if it is correct to indicate “(Figure 6A)” here.
Line 478: Maybe the correct figure is the “Figure1”?
Line 508: Please correct double brackets into single ones.
Line 524: Maybe should be “Figure 4” and not Figure 7?
Response: This was changed to read “Figure 4” as well pointed out by this reviewer.
Line 540: “notch” is reported without capital letter. Please correct.
Line 551: ONC affects, the “s” is missing.
Reviewer 2 Report
In the manuscript (ID: genes-775202) entitled “Optic nerve crush activates the genomic fabrics of the complement cascade and Delta-Notch signaling”, Victorino and colleagues profiled the transcriptome in a rat optic nerve crush (ONC) glaucoma model via Agilent gene expression microarrays. In detail, they applied comprehensive bioinformatic tools including STRING, KEGG/Pathvisio3 pathway and enrichment analysis, which demonstrated that complement activation as well as the Delta-Notch signaling pathway are mainly affected after ONC. Furthermore, gene expression of some dysregulated genes was validated by quantitative real-time PCR(RTq-PCR) analysis. Collectively, the manuscript provides novel and very interesting information about transcriptional changes 14 days after glaucomatous retina/retinal ganglion cell degeneration.
Despite interesting information about the transcriptome composition in the retina following ONC, the manuscript in its present form is not sufficient for publication in the Journal Genes. Concerns are listed as follows and arenot meant to discourage the authors but to improve the quality of the manuscript.
Major concerns:
- Gene regulation of the two highest upregulated genes Cd74and C3 was validated via RTq-PCR analysis. Tables S1 and S2 include the 50 most up- and the 50 most down-regulated genes, while Table 6 includes the top 50 up-regulated genes of the complement and Delta/Notch pathway (page 11, section 3.6). As the manuscript suggests that especially components of the classical and alternative complement system (including associated molecules of MAC) are up-regulated in their model (page 11, line 342-345), the authors should at least verify their dysregulation on mRNA level via RTq-PCR. Additionally, constituents of the Delta/Notch pathway should be validated by RTq-PCR analysis.
Minor concerns:
- Please check the Material and Method section and provide more experimental details for the reader (e.g. describe the dissection of the retina in the Material and Method section 2.4 (isolation for microarray analysis). Section 2.2: Which microscope was used? How much visual fields (peripheral/central) per retina were counted?
- In order to distinguish genes from proteins, all gene names should be written in italics. Please check the manuscript thoroughly (e.g. page 5, primer pairs; page 7, line 252, Cd74 and C3).
- The authors refer to supplementary Figures S1-S7 and Tables S1-S8 in the Result section. Please check their order in the text, e.g. Table S7 is listed in the Result section (page 7, line 254), but Table S5 is listed later (page 9, line 296).
- Please check the section Supplementary materials (page 19) for completeness. The authors should verify that all supplementary data have been listed/deposited.
- The authors provide bioinformatic data in regard to predicted protein-protein interaction networks. As they provide no evidence for changes on protein level, they should at least discuss the limitation of their study/a future perspective.
- To improve the language of the manuscript, a further proofreading should be done.
Author Response
In the manuscript (ID: genes-775202) entitled “Optic nerve crush activates the genomic fabrics of the complement cascade and Delta-Notch signaling”, Victorino and colleagues profiled the transcriptome in a rat optic nerve crush (ONC) glaucoma model via Agilent gene expression microarrays. In detail, they applied comprehensive bioinformatic tools including STRING, KEGG/Pathvisio3 pathway and enrichment analysis, which demonstrated that complement activation as well as the Delta-Notch signaling pathway are mainly affected after ONC. Furthermore, gene expression of some dysregulated genes was validated by quantitative real-time PCR(RTq-PCR) analysis. Collectively, the manuscript provides novel and very interesting information about transcriptional changes 14 days after glaucomatous retina/retinal ganglion cell degeneration.
Despite interesting information about the transcriptome composition in the retina following ONC, the manuscript in its present form is not sufficient for publication in the Journal Genes. Concerns are listed as follows and are not meant to discourage the authors but to improve the quality of the manuscript.
Major concerns:
Gene regulation of the two highest upregulated genes Cd74and C3 was validated via RTq-PCR analysis. Tables S1 and S2 include the 50 most up- and the 50 most down-regulated genes, while Table 6 includes the top 50 up-regulated genes of the complement and Delta/Notch pathway (page 11, section 3.6). As the manuscript suggests that especially components of the classical and alternative complement system (including associated molecules of MAC) are up-regulated in their model (page 11, line 342-345), the authors should at least verify their dysregulation on mRNA level via RTq-PCR. Additionally, constituents of the Delta/Notch pathway should be validated by RTq-PCR analysis.
Response: We recognize that it is important fo validate gene expression differences using independent methods, using such methods as RT-qPCR and immunodetection, and in numerous previous studies we have demonstrated correspondence using multiple methods (e.g., list a few here).
However, identical results are not expected because each method is subject to additional sources of technical noise (see Athanasiou et al., 2020) and the distinct samples required for these analyses introduces additional sources of biological variability.For these reasons we have chosen to validate expression differences that are not only highly significant but also represent high abundance transcripts, which are likely to overcome technical noise and biological variability.
- Athanasiou AT, Nussbaumer T, Kummer S, Hofer M, Johnston IG, Staltner M, Allmer DM, Scott MC, Vogl C, Fenger JM, Modiano JF, Walter I, Steinborn R. S100A4 mRNA-protein relationship uncovered by measurement noise reduction. J Mol Med (Berl). 2020 Apr 15. doi: 10.1007/s00109-020-01898-8.
Minor concerns:
Please check the Material and Method section and provide more experimental details for the reader (e.g. describe the dissection of the retina in the Material and Method section 2.4 (isolation for microarray analysis). Section 2.2: Which microscope was used? How much visual fields (peripheral/central) per retina were counted?
Response: These procedures were described in more details in the Methods section as suggested by this reviewer.
In order to distinguish genes from proteins, all gene names should be written in italics. Please check the manuscript thoroughly (e.g. page 5, primer pairs; page 7, line 252, Cd74 and C3).
Response: The gene names were now written in italics throughout the manuscript.
The authors refer to supplementary Figures S1-S7 and Tables S1-S8 in the Result section. Please check their order in the text, e.g. Table S7 is listed in the Result section (page 7, line 254), but Table S5 is listed later (page 9, line 296).
Response: Numbering of figures and tables was revised as well as their citations in the text, thank you.
Please check the section Supplementary materials (page 19) for completeness. The authors should verify that all supplementary data have been listed/deposited.
Response: All supplementary materials were checked and are now in accordance with what is described in the text.
The authors provide bioinformatic data in regard to predicted protein-protein interaction networks. As they provide no evidence for changes on protein level, they should at least discuss the limitation of their study/a future perspective.
Response: This is an interesting point raised by the reviewer and we included a sentence in the discussion section (page 20, lines 496-499).
To improve the language of the manuscript, a further proofreading should be done.
Response: We agree with this reviewer and have made a thorough proofreading of the manuscript in order to improve readability.
Round 2
Reviewer 1 Report
Reviewer’s Comments (genes-775202)
Title: Optic nerve crush activates the genomic fabrics of the Complement cascade and DeltaNotch signalling
Even though the major and minor issues previously reported were corrected, the manuscript is suitable for publication after the authors will address some minor revisions outlined below:
Minor Revisions:
- Please insert a bracket before "Figure 8" in line 459
- Although OPC has been corrected in the text, it should indeed be corrected in all the other passages that include the same error (caption of the formulas, Line 170, 185 etc).
- Please delete “gene expression stability score”
- Please delete “stability of expression” in line 357 and use only the relative abbreviation
- Please correct “(Figure 1D)” with (Figure 1d) in line 536
- Please change “complement cascade” with “Complement Cascade” (capital letters) in line 589
Furthermore, in the supplementary materials:
- Figure S1 is missing.
- The figure legend of Figure S1 reported "Figure 1". Please correct.
- Please correctly realign the numbers reported in Tables S1 and S3.
Author Response
Even though the major and minor issues previously reported were corrected, the manuscript is suitable for publication after the authors will address some minor revisions outlined below:
Response: We thank the reviewers for the positive feedback and the suggestions, that improved our manuscript. All minor revisions were corrected in the R2 version of the manuscript and are indicated in yellow in the text.
Minor Revisions:
- Please insert a bracket before "Figure 8" in line 459
Response: This was corrected, thank you.
- Although OPC has been corrected in the text, it should indeed be corrected in all the other passages that include the same error (caption of the formulas, Line 170, 185 etc).
Response: OPC was now changed to ONC throughout the text and formulas as indicated by this reviewer.
- Please delete “gene expression stability score”
- Please delete “stability of expression” in line 357 and use only the relative abbreviation
- Please correct “(Figure 1D)” with (Figure 1d) in line 536
- Please change “complement cascade” with “Complement Cascade” (capital letters) in line 589
Response: These changes were made as requested in the revised manuscript.
Furthermore, in the supplementary materials:
- Figure S1 is missing
- The figure legend of Figure S1 reported "Figure 1". Please correct.
- Please correctly realign the numbers reported in Tables S1 and S3.
Response: Figure S1 was mislabeled as “Appendix 1”. The references to this figure in page 35 were now corrected. Tables were also realigned as requested.
